# Bi-Lipschitz Autoencoder With Injectivity Guarantee

**Qipeng Zhan**[*], **Zhuoping Zhou**[*], **Zexuan Wang**[*], **Qi Long**[†], **Li Shen**[†]
University of Pennsylvania
{qipengz, zhuopinz, zxwang}@sas.upenn.edu
{qlong@, li.shen@pennmedicine.}upenn.edu
[*]Equal contribution. [†]Corresponding authors

## ABSTRACT

Autoencoders are widely used for dimensionality reduction, based on the assumption that high-dimensional data lies on low-dimensional manifolds. Regularized autoencoders aim to preserve manifold geometry during dimensionality reduction, but existing approaches often suffer from non-injective mappings and overly rigid constraints that limit their effectiveness and robustness. In this work, we identify encoder non-injectivity as a core bottleneck that leads to poor convergence and distorted latent representations. To ensure robustness across data distributions, we formalize the concept of admissible regularization and provide sufficient conditions for its satisfaction. In this work, we propose the Bi-Lipschitz Autoencoder (BLAE), which introduces two key innovations: (1) an injective regularization scheme based on a separation criterion to eliminate pathological local minima, and (2) a bi-Lipschitz relaxation that preserves geometry and exhibits robustness to data distribution drift. Empirical results on diverse datasets show that BLAE consistently outperforms existing methods in preserving manifold structure while remaining resilient to sampling sparsity and distribution shifts. Code is available at https://github.com/qipengz/BLAE.

## 1 INTRODUCTION

Autoencoders have been established as powerful tools for dimensionality reduction and visualization. While theoretically promising due to their universal approximation capabilities, vanilla autoencoders often fail to preserve the geometric properties essential for meaningful latent representations. This limitation has motivated two main approaches to regularize the latent space: 1) Gradient-based methods (Nazari et al., 2023; Salah et al., 2011; Lim et al., 2024a; Lee et al., 2022) constrain the Jacobian of the encoder or decoder to promote smoothness and local geometric preservation. 2) Graph-based methods align latent representations with distance structures derived from neighborhood graphs (Moor et al., 2020; Schönenberger et al., 2020; Singh & Nag, 2021; Zhan et al., 2025) or pretrained embeddings (Mishne et al., 2019; Duque et al., 2022).

Despite their distinct theoretical foundations, both approaches face substantial practical challenges. Graph-based methods depend heavily on the graph accuracy, limiting their effectiveness under sparse sampling conditions. Gradient-based methods exhibit robustness to sample size and yield smoother manifolds but frequently converge to local minima during optimization, resulting in compromised performance that falls short of theoretical expectations.

From a differential topology perspective, gradient constraints typically encode local geometric properties. For example, a mapping $f : \mathcal{M} \to \mathbb{R}^n$ satisfying $J_f^\top J_f \equiv I$ is an isometric immersion. With an additional global injectivity condition, $f$ becomes an embedding. This motivates us to investigate the interplay between injectivity and optimization in gradient-based autoencoders. Through theoretical and empirical analysis, we identify the non-injectivity of the encoder as a key bottleneck in training autoencoders effectively. To address this limitation, we propose a novel framework that guarantees injectivity through separation criteria in the latent space, helping to avoid pathological local minima that can trap standard gradient-based methods and enabling them to achieve their theoretical potential.

To create a geometrically consistent latent embedding that maintains intrinsic manifold structures, robustness against distributional shifts is essential because our goal isn't tied to specific data distributions. This requires the imposed regularization to be admissible — specifically, the targeted geometric properties must be strictly satisfied. While prior work employs isometric constraints (Lee et al., 2022; Gropp et al., 2020; Lim et al., 2024a), such embeddings typically demand $\mathcal{O}(m^2)$ dimensions, where $m$ is the dimension of the manifold. This severely impairs the efficiency of the autoencoder. To address this, we propose a principled relaxation through bi-Lipschitz regularization, achieving linear complexity in manifold dimension while maintaining robustness to distribution shifts.

To validate our theoretical insights, we develop the Bi-Lipschitz Autoencoder (BLAE), which combines injective and bi-Lipschitz regularization. Our experiments across multiple datasets demonstrate that BLAE preserves manifold structure with higher fidelity than existing methods while exhibiting significant robustness to distribution shifts and sparse sampling conditions.

The rest of this paper is organized as follows. Section 2 discusses the limitations of existing methods. Section 3 introduces our proposed framework. Section 4 reviews related work. Section 5 presents experimental results, and Section 6 concludes the paper.

## 2 BOTTLENECK OF AUTOENCODERS

Throughout this work, we assume the intrinsic data manifold $\mathcal{M} \subset \mathbb{R}^m$ is a compact and connected Riemannian manifold with Riemannian measure $\mu_{\mathcal{M}}$, unless otherwise stated. We also assume the data distribution $\mathbb{P}$ on $\mathcal{M}$ is equivalent to $\mu_{\mathcal{M}}$, i.e., $\mathbb{P} \ll \mu_{\mathcal{M}}$ and $\mu_{\mathcal{M}} \ll \mathbb{P}$.

### 2.1 AUTOENCODERS AND REGULARIZATIONS

Conventional manifold learning and dimension reduction frameworks typically assume that high-dimensional data resides on a low-dimensional manifold $\mathcal{M} \subset \mathbb{R}^m$. An autoencoder learns this intrinsic representation through a pair of parameterized mappings $(\mathcal{E}_\theta, \mathcal{D}_\phi)$ via neural networks:

- The encoder $\mathcal{E}_\theta : \mathbb{R}^m \to \mathbb{R}^n (n \ll m)$ compresses data onto a low-dimensional latent space;
- The decoder $\mathcal{D}_\phi : \mathbb{R}^n \to \mathbb{R}^m$ reconstructs the original data from the latent representation.

Training optimizes this encoder-decoder pair by minimizing the expected reconstruction error:

$$\mathcal{L}_{\text{recon}} = \mathbb{E}_{x \sim \mathbb{P}} \big\| x - \mathcal{D}_\phi(\mathcal{E}_\theta(x)) \big\|^2. \tag{1}$$

While vanilla autoencoders effectively learn compressed representations, they often fail to preserve essential data properties. Advanced variants have emerged to address requirements such as representation smoothness, model robustness, and geometric preservation. Based on their regularization mechanisms, we classify these variants into three categories.

**Gradient-regularization.** This category applies explicit constraints on the derivatives of network mappings. Formally, gradient regularization is expressed as:

$$\mathcal{L}_{\text{grad}} = \mathbb{E}_{x \sim \mathbb{P}} \big[ R_1(J_f(x)) \big], \tag{2}$$

where $R_1$ is a loss function, and $J_f(x)$ represents the Jacobian matrix of a differentiable function $f$ at input $x$. The function $f$ can be instantiated as either the encoder $\mathcal{E}_\theta$ or the decoder $\mathcal{D}_\phi$. A canonical example is the contractive autoencoder (Salah et al., 2011), where $R_1(\cdot) = \| \cdot \|_F^2$ imposes Frobenius-norm constraints on the encoder's Jacobian to promote local stability.

**Geometry-regularization.** This category preserves distance relationships when mapping from the manifold $\mathcal{M}$ to the latent space $\mathcal{N}$. Geometry regularization takes the form:

$$\mathcal{L}_{\text{geo}} = \mathbb{E}_{x,y \sim \mathbb{P}} \left[ R_2(d_{\mathcal{M}}(x,y), d_{\mathcal{N}}(\mathcal{E}_\theta(x), \mathcal{E}_\theta(y))) \right], \tag{3}$$

where $R_2$ is a loss function and $d_*$ denotes the geodesic distance on the corresponding manifold. Implementation typically involves approximating geodesic distances $d_{\mathcal{M}}$ through $k$-nearest neighbors ($k$-NN) graph construction and shortest-path computations. The latent space metric $d_{\mathcal{N}}$ is conventionally defined as the Euclidean distance $\| \cdot \|$.

**Embedding-regularization.** The last category directly guides the encoder to match embeddings produced by classical dimension reduction methods. The regularization is formulated as:

$$\mathcal{L}_{\text{emb}} = \mathbb{E}_{x \sim \mathbb{P}} \| \mathcal{E}_\theta(x) - z \|^2, \tag{4}$$

where $z$ is the target embedding of $x$ obtained from methods such as Isomap or diffusion maps.

**Remark 1.** Since both embedding and geometry regularizations rely on graph construction procedures standard in conventional dimensionality reduction methods, we unify them under the framework of **Graph-Regularization**.

## 2.2 WHY AUTOENCODERS AND GRADIENT VARIANTS FAIL?

The universal approximation theorem grants neural network-based autoencoders significant potential for learning effective latent representations. However, in practice, standard gradient descent optimization frequently converges to poor local minima rather than discovering globally optimal solutions. This convergence behavior—whether terminating in suboptimal local minima or achieving global optimality—is fundamentally connected to the concept of *injection*.

**Definition 1** (Injection). $f$ is an injection (injective mapping) on $\mathcal{M}$, if $f(x) \neq f(y), \forall x \neq y \in \mathcal{M}$.

Non-injective encoders create latent space collisions where distinct data points map to identical or nearly identical codes, resulting in inevitable reconstruction errors and suboptimal model performance[1]. In practical implementations with finite sample sets $\{x_1, \cdots, x_N\} \subset \mathcal{M}$, the point-wise injectivity condition $\mathcal{E}_\theta(x_i) \neq \mathcal{E}_\theta(x_j)$ for all $i \neq j$ typically holds ($w.p.1$) when the encoder $\mathcal{E}_\theta$ avoids local constancy. However, this does not guarantee global injectivity. More precisely, even when $\mathcal{E}_\theta(x_i) \neq \mathcal{E}_\theta(x_j)$ for distinct sample points, if disjoint neighborhoods $U_i, U_j$ of these points have overlapping encodes ($\mathcal{E}_\theta(U_i) \cap \mathcal{E}_\theta(U_j) \neq \varnothing$), the injectivity property is violated.

Figure 1 illustrates this non-injective bottleneck using a toy example of 20 points sampled from a V-shaped manifold. We analyze three autoencoder configurations with two hidden layers of varying capacities (hidden dimensions: 2, 16, 256). The condition $U_i \cap U_j = \varnothing$ and $\mathcal{E}_\theta(U_i) \cap \mathcal{E}_\theta(U_j) \neq \varnothing$ manifests when the encoder maps points from distant manifold regions to nearby coordinates in the latent space. Figures 1 (h) (j) (l) demonstrate this distortion between red and blue classes.

To compensate for these latent space collisions, the decoder must generate sharp variations, appearing as high local curvature, within intersection regions to minimize reconstruction error, as shown in Figure 1 (k). The required network complexity for these variations scales polynomially with the density of encoded data points. When the decoder's capacity cannot accommodate these geometric demands, the optimization process becomes trapped in suboptimal local minima where gradient signals align with the network's expressivity boundaries (Figure 1 (g)).

Notably, gradient-based regularization schemes suffer from a similar injectivity bottleneck. While these methods effectively constrain local mapping properties through differential constraints, they remain insufficient for ensuring global injectivity—a critical topological property that distinguishes proper embeddings from mere immersions. In approaches like (Lim et al., 2024a; Lee et al., 2022), isometric constraints yield locally structure-preserving encoders via isometric immersion but fail to satisfy the additional topological requirements for genuine manifold embedding.

## 2.3 GRAPH REGULARIZATIONS VS. GRADIENT REGULARIZATIONS

Unlike gradient-based approaches, graph-based autoencoder variants inherently enforce injective mapping by requiring distance preservation between manifolds $\mathcal{M}$ and latent space $\mathcal{N}$, where $d_\mathcal{M} > 0 \Rightarrow d_\mathcal{N} > 0$. However, these graph-driven approaches have two fundamental limitations: i) Their discrete graph approximations become unreliable under sparse sampling conditions, as shortest-path distances increasingly deviate from true manifold geodesics; ii) The Euclidean metric assumption in latent space introduces systematic errors unless $\mathcal{N}$ satisfies strict convexity requirements. Comparatively, when the injectivity of the encoder is guaranteed, gradient-based methods can achieve smoother embeddings with greater robustness to variations in sample density.

---

[1]Formally, as the reconstruction loss is calculated in expectation, injective guarantees need only hold a.e..

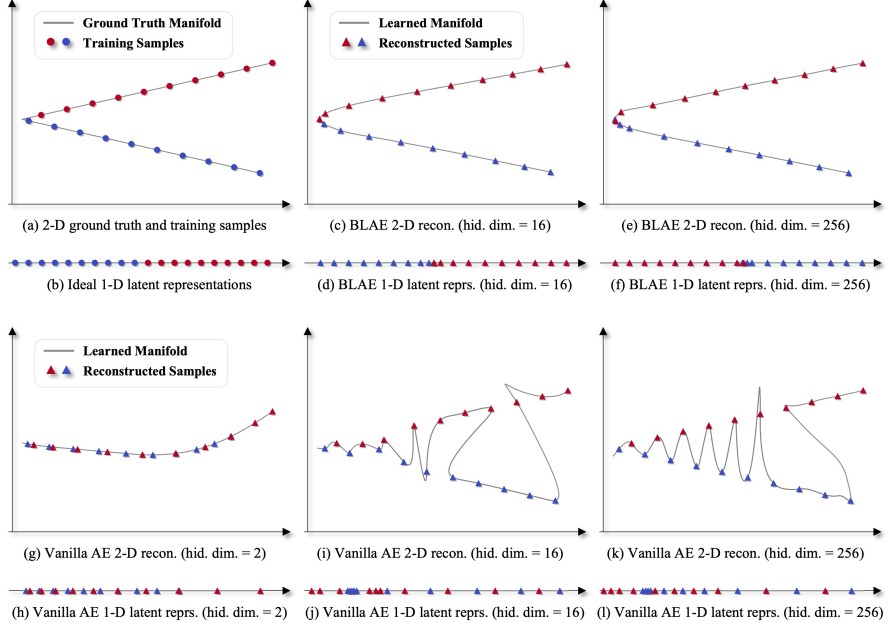

Figure 1: Toy example demonstrating the non-injective encoder bottleneck. (a) 20 training points sampled from a V-shaped manifold with two classes (red and blue). (b) Ground truth: an ideal encoder separates the two classes in a 1D latent space. (c) (e) Reconstructed manifolds by BLAE (ours) with hidden dimensions 16 and 256, respectively. (d) (f) Corresponding latent representations learned by BLAE showing proper class separation. (g) (i) (k) Reconstructed manifolds by vanilla autoencoders with hidden dimensions 2, 16, and 256. (h) (j) (l) Corresponding latent representations showing non-injective collapse, where distant manifold regions (red vs. blue) map to overlapping latent codes, resulting in pathological local minima.

## 3 GRADIENT AUTOENCODER WITH INJECTIVE CONSTRAINT

### 3.1 SEPARATION CRITERION AND INJECTIVE REGULARIZATION

Our theoretical analysis in Section 2.2 yields two critical insights: i) Imposing injectivity constraints on the encoder can effectively eliminate pathological local minima that trap optimization trajectories. ii) Sample-level condition $\mathcal{E}_\theta(x_i) \neq \mathcal{E}_\theta(x_j)$ is insufficient for global injectivity. To address these issues, we must prevent distant manifold neighborhoods from collapsing into proximal latent clusters, leading us to enforce metric *separation*:

**Definition 2** ($(\delta, \epsilon)$-separation). Given $\delta, \epsilon > 0$, a mapping $f : \mathcal{M} \to \mathcal{N}$ is $(\delta, \epsilon)$-separated if for all $x, y \in \mathcal{M}$ satisfying $d_\mathcal{M}(x, y) \geq \delta$:

$$\frac{d_\mathcal{N}(f(x), f(y))}{d_\mathcal{M}(x, y)} > \epsilon. \tag{5}$$

This separation criterion provides a sufficient condition for $f$ to be injective: if $f$ is $(\delta, \epsilon)$-separated for any $\delta, \epsilon > 0$, then $f$ must be an injection. Indeed, under some mild assumptions, this condition serves as an equivalent characterization of injection:

**Theorem 1.** [2] *Suppose $f : \mathcal{M} \to \mathcal{N}$ is continuous and $\mathcal{M}$ is compact, then $f$ is injective if and only if for any $\delta > 0$, there exists $\epsilon > 0$ such that $f$ is $(\delta, \epsilon)$-separated.*

To address the limitations of naive injectivity constraints, we propose a regularization that penalizes sample pairs violating the separation condition:

$$\mathcal{L}_{\text{inj}}(\delta, \epsilon) = \mathbb{E}_{x,y \sim \mathbb{P}}\left[\text{ReLU}\left(\log \frac{\epsilon d_\mathcal{M}(x, y)}{d_\mathcal{N}(\mathcal{E}_\theta(x), \mathcal{E}_\theta(y))}\right) \cdot \mathbf{1}_{d_\mathcal{M}(x,y) > \delta}\right]. \tag{6}$$

---

[2]All proofs are deferred to Appendix A.

However, this penalty permits a trivial optimization path to zero loss: simply scaling the encoder by a factor $k = \epsilon \cdot \max \frac{d_{\mathcal{M}}(x_i, x_j)}{d_{\mathcal{N}}(\mathcal{E}_\theta(x_i), \mathcal{E}_\theta(x_j))}$. To prevent this, we additionally constrain the encoder to be *non-expansive*, meaning $\forall x, y \in \mathcal{M}, d_{\mathcal{N}}(\mathcal{E}_\theta(x), \mathcal{E}_\theta(y)) \leq d_{\mathcal{M}}(x, y)$. Our final regularization term combines both constraints:

$$\mathcal{L}_{\text{reg}}(\delta, \epsilon) = \mathcal{L}_{\text{inj}}(\delta, \epsilon) + \alpha \cdot \mathbb{E}_{x, y \sim \mathbb{P}} \left[ \text{ReLU} \left( \frac{d_{\mathcal{N}}(\mathcal{E}_\theta(x), \mathcal{E}_\theta(y))}{d_{\mathcal{M}}(x, y)} - 1 \right) \cdot \mathbf{1}_{d_{\mathcal{M}}(x, y) > \delta} \right], \quad (7)$$

where $\alpha$ (default: 5) is a weighting factor calibrating the strength of the non-expansive constraint.

**Remark 2.** While Theorem 1 requires validating the separation condition for all $\delta > 0$, in practical implementations, it suffices to validate at a threshold $\delta_{\min} = \min_{i \neq j} d_{\mathcal{M}}(x_i, x_j)$.

Similar to graph-based approaches, we use Euclidean distance for $d_{\mathcal{N}}$ and approximate $d_{\mathcal{M}}$ through graph construction. Although this approximation introduces some systematic error as discussed in Section 2.3, our separation criterion proves remarkably resilient to such approximations compared to other geometry-based regularizations. This robustness allows effective combination with gradient regularization techniques (see Appendix C.4 for details).

The proposed injective regularization systematically mitigates pathological local minima in the loss landscape. Figure 2 visualizes this effect through 2D loss landscapes comparing a standard autoencoder with its injective-regularized counterpart. Both models were initialized at $\theta_0$ and optimized to $\theta_1$ (vanilla autoencoder) and $\theta_2$ (our regularized autoencoder), respectively. The contour maps represent loss values across the parameter subspace spanned by vectors $\theta_0 - \theta_1$ and $\theta_0 - \theta_2$. Figure 2(a) reveals how the reconstruction loss landscape contains a local minima that trap the vanilla model during optimization. In contrast, Figure 2(b) illustrates how our combined loss (reconstruction plus injective regularization) reshapes the landscape to provide smoother optimization paths toward the superior global minima.

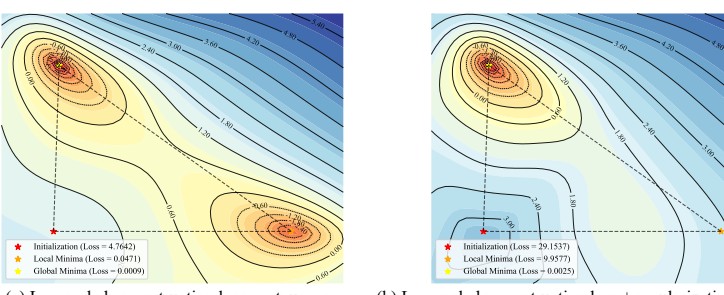

(a) Log-scaled reconstruction loss contour map      (b) Log-scaled reconstruction loss + regularization contour map

Figure 2: Loss landscapes of autoencoders on Swiss roll data. Warmer colors indicate lower loss. (a) Log-scale contour of reconstruction loss, showing the presence of local minima. (b) Log-scale contour of reconstruction loss combined with regularization, where local minima are eliminated.

## 3.2 ROBUSTNESS TO DISTRIBUTION SHIFT AND ADMISSIBLE REGULARIZATION

When training autoencoders, both reconstruction error and regularization terms are computed as expectations over a specific data distribution $\mathbb{P}$ on $\mathcal{M}$. This typically causes the resulting latent space embedding $\mathcal{N}$ to be influenced by $\mathbb{P}$. However, our fundamental goal is to learn a low-dimensional embedding of the manifold structure itself, independent of any particular data sampling distribution. To achieve robustness against distribution shifts, we introduce the concept of *Admissibility*:

**Definition 3** (Admissibility). For a regularization term $\mathbb{E}_{x \sim \mathbb{P}}[R(f_\Theta(x))]$, where $R$ is a loss function and $f_\Theta$ belongs to a parameterized smooth function class $\mathcal{F}_\Theta$ (which may represent the encoder, decoder, or their Jacobians), let

$$S_{\mathbb{P}} := \arg \min_{f_\Theta \in \mathcal{F}_\Theta} \mathbb{E}_{x \sim \mathbb{P}}[R(f_\Theta(x))]. \quad (8)$$

The regularization is admissible if for any probability measures $\mathbb{P}$ and $\mathbb{Q}$ which are both equivalent to $\mu_{\mathcal{M}}$, $S_{\mathbb{P}} = S_{\mathbb{Q}}$, i.e., the set of global minima is independent of the probability measure.

The following theorem provides a constructive approach for designing admissible regularizations:

**Theorem 2.** *Let $R$ be a loss function that has a global minimum, and if*

$$\min_{f_\Theta \in \mathcal{F}_\Theta} \mathbb{E}_{x \sim \mathbb{P}}[R(f_\Theta(x))] = \min_u R(u), \tag{9}$$

*then the corresponding regularization $\mathbb{E}_{x \sim \mathbb{P}}[R(f_\Theta(x))]$ is admissible.*

Admissible regularizations typically enforce specific geometric or functional properties uniformly across the manifold. Consider, for example, a regularizer of the form $R(f(x)) = \|(f(x))^\top f(x) - I\|_F^2$, where $f$ represents the Jacobian of the decoder. This formulation imposes an isometric constraint, and achieving its minimum ensures the decoder maintains local isometry at each input point $x$. The admissibility of such a regularization guarantees that this isometric property is preserved regardless of how data points are sampled from the manifold.

**Remark 3.** The standard reconstruction error can itself be viewed as a special case of admissible regularization, where $R = \| \cdot \|^2$ and $\mathcal{F}_\Theta = \{\mathcal{D}_\theta \circ \mathcal{E}_\phi - \mathrm{id} \mid \theta, \phi\}$, with 'id' representing the identity mapping. This reconstruction term is inherently admissible.

## 3.3 BI-LIPSCHITZ RELAXATION

The injectivity property enables the integration of complementary gradient regularizations into autoencoders, enhancing smoothness and geometric fidelity. For robustness against distribution shifts, we need these gradient regularizations to be admissible. Previous research has explored isometric constraints for geometric preservation, but these formulations are proven overly restrictive in practice.

According to the Nash embedding theorem (Nash, 1956), an $k$-dimensional compact Riemannian manifold requires a latent dimension of $\mathcal{O}(k^2)$ to guarantee an isometric embedding[3]. This quadratic scaling contradicts the fundamental purpose of autoencoders as tools for efficient dimensionality reduction. For example, even a modest manifold of intrinsic dimension 10 would theoretically require over 50 dimensions for isometric embedding, introducing substantial representational redundancy.

Furthermore, when the latent dimension fails to meet the requirements for isometric embedding, the corresponding regularization becomes non-admissible, which motivates us to introduce bi-Lipschitz regularization, a principled relaxation scheme that balances geometric preservation with admissibility.

**Definition 4** (Bi-Lipschitz). A mapping $f : \mathcal{M} \to \mathcal{N}$ is $\kappa$-Bi-Lipschitz, where $\kappa \geq 1$, if

$$\frac{1}{\kappa} \cdot d_\mathcal{M}(x, y) \leq d_\mathcal{N}(f(x), f(y)) \leq \kappa \cdot d_\mathcal{M}(x, y), \quad \forall x, y \in \mathcal{M}. \tag{10}$$

While geodesic distance approximation introduces estimation errors, we can establish the bi-Lipschitz property through differential analysis using the gradient of $f$. Let $f_\mathcal{M}$ denote $f$ restricted to $\mathcal{M}$, and $J_f^\mathcal{M}(x)$ be the Jacobian of $f$ restricted to the tangent space $T_x\mathcal{M}$. We then have:

**Theorem 3.** *Let $f : \mathcal{M} \to \mathbb{R}^n$ be a smooth mapping. If $\mathcal{M}$ is connected and $f$ is a diffeomorphism, then $f$ is $\kappa$-bi-Lipschitz if and only if*

$$\frac{1}{\kappa} \leq \sigma_{\min}(J_f^\mathcal{M}(x)) \leq \sigma_{\max}(J_f^\mathcal{M}(x)) \leq \kappa, \quad \forall x \in \mathcal{M} \tag{11}$$

*where $\sigma_{\min}(J_f^\mathcal{M}(x))$, $\sigma_{\max}(J_f^\mathcal{M}(x))$ are the minimum and maximum singular values of $J_f^\mathcal{M}(x)$.*

Theorem 3 provides a principled approach to rigorously analyze bi-Lipschitz properties without requiring explicit geodesic distance computation. However, computing singular values of $J_f^\mathcal{M}(x)$ requires knowledge of the tangent space, and existing methods (Lim et al., 2024b; Zhang & Zha, 2003) for estimating tangent spaces from empirical data introduce additional errors.

Two key insights help address these challenges: i) When $f$ is bi-Lipschitz on $\mathbb{R}^m$, it is naturally bi-Lipschitz on any sub-manifold $\mathcal{M}$. This allows substituting $\mathcal{M}$ with $\mathbb{R}^m$ in Theorem 3. ii) When $m > n$ (as in typical encoding scenarios), $f$ cannot be a diffeomorphism due to dimensional incompatibility. Therefore, we apply condition equation 11 to the decoder $\mathcal{D}_\phi : \mathbb{R}^n \to \mathbb{R}^m$ rather than the encoder $\mathcal{E}_\theta : \mathbb{R}^m \to \mathbb{R}^n$, since $f$ is $\kappa$-bi-Lipschitz if and only if $f^{-1}$ is $\kappa$-bi-Lipschitz.

---

[3]The required latent dimension $n$ satisfies $\frac{k(k+1)}{2} \leq n \leq \frac{k(3k+11)}{2}$.

Our proposed bi-Lipschitz regularization is formulated as:

$$\mathcal{L}_{\text{bi-Lip}}(\kappa) = \mathbb{E}_{x \sim \mathbb{P}}\left[\text{ReLU}(\frac{1}{\kappa} - \sigma_{\min}(x))^2 + \text{ReLU}(\sigma_{\max}(x) - \kappa)^2\right], \quad (12)$$

where $\sigma_{\min}(x)$ and $\sigma_{\max}(x)$ represent the smallest and largest singular values of $J_{\mathcal{D}_\phi}(x)$, respectively. This regularization retains admissibility even with relatively low embedding dimensions ($\mathcal{O}(m)$):

**Theorem 4.** *Suppose $\mathcal{M} \subset \mathbb{R}^m$ is a connected compact $k$-dimensional Riemannian manifold. Then there exists a $\kappa$-bi-Lipschitz mapping that embeds $\mathcal{M}$ into $\mathbb{R}^n$ for some $\kappa \geq 1$ and $k \leq n \leq 2k$.*

When the manifold $\mathcal{M}$ admits a $\kappa$-bi-Lipschitz embedding in $\mathbb{R}^n$, the corresponding loss function achieves its minimum (zero) and hence satisfies the assumption of Theorem 2. Consequently, the bi-Lipschitz regularization is admissible.

We refer to an autoencoder regularized by both bi-Lipschitz and injective constraints as a **Bi-Lipschitz Autoencoder (BLAE)**:

$$\mathcal{L}_{\text{BLAE}} = \mathcal{L}_{\text{recon}} + \lambda_{\text{reg}} \cdot \mathcal{L}_{\text{reg}} + \lambda_{\text{bi-Lip}} \cdot \mathcal{L}_{\text{bi-Lip}}, \quad (\text{BLAE})$$

where $\lambda_{\text{reg}}$ and $\lambda_{\text{bi-Lip}}$ are weighting factors. The injective term eliminates local minima caused by non-injective encoders, while the bi-Lipschitz term ensures consistent geometric mapping regardless of data distribution. Together, these constraints enable BLAE to preserve manifold structure with higher fidelity and robustness than existing methods.

**Remark 4.** While Theorem 3 assumes smoothness (satisfied by neural networks using activation functions like tanh, sigmoid, ELU), our framework naturally extends to continuous piecewise-smooth mappings. This preserves the theoretical conclusions even with non-smooth activation functions like ReLU, demonstrating the universal applicability of our results across neural network architectures.

## 4 RELATED WORK

Standard autoencoders often exhibit geometric distortion in the latent space. To address this, regularized methods have been proposed to preserve geometric structure, which can be grouped into three main paradigms based on their regularization mechanisms:

**Embedding-regularized autoencoder.** Conventional dimensionality reduction techniques such as ISOMAP (Tenenbaum et al., 2000), LLE (Roweis & Saul, 2000), t-SNE (Van der Maaten & Hinton, 2008), and UMAP (McInnes et al., 2018) effectively preserve geometric structures through neighborhood graphs. However, these methods lack explicit mappings between the original and latent spaces, limiting their ability to generalize to new data points. To overcome this, Duque et al. (2022) introduced the Geometry Regularized Autoencoder, a unified framework that integrates autoencoders with classical dimensionality reduction methods to enable robust extension to unseen data. Similarly, Diffusion Nets (Mishne et al., 2019) enhances autoencoders by learning embedding geometry from Diffusion Maps (Coifman & Lafon, 2006) through additional eigenvector constraints.

**Geometry-regularized autoencoder.** This approach enforces geometric properties in the latent space via graph-based regularizations. Neighborhood Reconstructing Autoencoders (Lee et al., 2021) reduce overfitting and connectivity errors by enforcing correctly reconstructed neighborhoods. Structure-Preserving Autoencoders (Singh & Nag, 2021) maintain consistent distance ratios between ambient and latent spaces. Topological Autoencoders (Moor et al., 2020) capture data's topological signature through homology groups. While initially developed for Euclidean spaces, these methods can be adapted to non-Euclidean manifolds by constructing neighborhood graphs and computing geodesic distances through shortest paths.

**Gradient-regularized autoencoder.** This paradigm directly constrains on the derivatives of the network mappings. Contractive Autoencoder (Salah et al., 2011) penalizes the Frobenius norm of the encoder's Jacobian matrix, enforcing local stability and enhancing feature robustness. Chen et al. (2020) proposed regularizing the decoder's induced metric tensor to learn flat manifold representations. Geometric Autoencoders (Nazari et al., 2023) preserve volume form in latent space by regularizing the determinant of the decoder's metric tensor, improving data visualization.

Recent advances focus on learning isometric embeddings: Gropp et al. (2020) enforces identity metric tensors stochastically, while Lee et al. (2022) achieves coordinate invariance through spectral

constraints. Graph Geometry-Preserving Autoencoders (Lim et al., 2024a) bridge graph-based and gradient-based approaches by leveraging graph Laplacian spectral properties to approximate the underlying Riemannian metric.

## 5 EXPERIMENTS

**Experimental setup.** We evaluate our approach against nine baselines: (1) geometry-based autoencoders (SPAE (Singh & Nag, 2021), TAE (Moor et al., 2020)), (2) gradient-based autoencoders (IRAE (Lee et al., 2022), GAE (Nazari et al., 2023), CAE (Salah et al., 2011)), (3) embedding-based autoencoders (GRAE (Duque et al., 2022), Diffusion Net (DN) (Mishne et al., 2019)), (4) a hybrid approach combining graph and gradient regularization (GGAE (Lim et al., 2024a)), and (5) a vanilla autoencoder. We conduct a grid search over hyperparameters for each model-dataset combination and report the best performance. Detailed implementation settings are provided in Appendix B.1.

**Evaluation metric.** We assess model performance through both reconstruction accuracy and geometric preservation. Reconstruction fidelity is quantified by Mean Squared Error (MSE) between original samples and reconstructions. Geometric preservation is evaluated using two metrics: (1) $k$-NN recall (Sainburg et al., 2021; Kobak et al., 2019), measuring neighborhood correspondence between latent and original spaces, and (2) $KL_\sigma$ divergence (Chazal et al., 2011) with bandwidths $\sigma \in \{0.01, 0.1, 1\}$, assessing similarity of distance distributions across scales. To better evaluate manifold structures, we adapt these metrics to use geodesic distances derived from similarity graphs rather than the original Euclidean formulations. See Appendix B.2 for details.

Table 1 reports the average ranks of all methods across metrics and datasets, providing a compact overview of overall performance that complements the per-dataset comparisons. BLAE achieves the highest average ranking on key metrics, reflecting superior performance in graph geometry preservation, reconstruction fidelity, and downstream task accuracy.

Table 1: Average ranks of evaluation metrics across all datasets (lower is better). Detailed metric values for each dataset are provided in Appendix B.4 and Appendix C.6. The best is shown in bold.

| Measure | BLAE | SPAE | TAE | DN | GRAE | CAE | GGAE | IRAE | GAE | Vanilla AE |
|---|---|---|---|---|---|---|---|---|---|---|
| $k$-NN | **1.8 ± 1.3** | 3.2 ± 1.3 | 3.8 ± 0.8 | 4.5 ± 2.7 | 4.0 ± 2.7 | 5.8 ± 1.8 | 7.5 ± 2.1 | 7.2 ± 0.4 | 9.0 ± 1.7 | 7.8 ± 0.8 |
| $KL_{0.01}$ | **1.0 ± 0.0** | 3.0 ± 0.0 | 2.5 ± 0.9 | 6.0 ± 1.2 | 4.5 ± 1.5 | 7.0 ± 0.7 | 8.2 ± 2.5 | 6.8 ± 2.4 | 6.5 ± 1.1 | 9.2 ± 0.4 |
| $KL_{0.1}$ | **1.0 ± 0.0** | 3.2 ± 1.1 | 2.8 ± 0.8 | 5.5 ± 2.2 | 3.5 ± 0.9 | 7.5 ± 0.9 | 9.0 ± 1.7 | 7.2 ± 1.6 | 6.5 ± 0.9 | 8.8 ± 0.4 |
| $KL_1$ | **1.0 ± 0.0** | 4.2 ± 2.3 | 3.2 ± 1.3 | 5.2 ± 2.8 | 4.2 ± 1.9 | 7.0 ± 1.6 | 8.0 ± 1.6 | 7.2 ± 1.3 | 6.0 ± 2.2 | 8.2 ± 0.8 |
| MSE | **1.2 ± 0.4** | 4.8 ± 3.3 | 4.0 ± 0.7 | 5.2 ± 3.1 | 4.5 ± 3.0 | 5.5 ± 1.5 | 7.5 ± 2.1 | 7.0 ± 2.1 | 8.0 ± 1.6 | 7.2 ± 1.3 |
| Accuracy | **1** | 5 | 7 | 4 | 6 | 2 | 10 | 8 | 3 | 9 |

**Swiss Roll.** The Swiss Roll dataset consists of a synthetic 2-dimensional manifold embedded in $\mathbb{R}^3$. We construct it by uniformly sampling points from $[-2, 10] \times [0, 6]$ and isometrically mapping them to $\mathbb{R}^3$. To create a meaningful contrast between Euclidean and geodesic distances, we remove a strip $[1.5, 6.5] \times [2.5, 3.5]$ from the data (see Appendix B.3 for details).

As shown in Figure 3, our BLAE method correctly preserves the geometric structure in latent space. Graph-based architectures (SPAE, TAE, GRAE, DN) successfully unroll the manifold but distort its geometry due to discrepancies between geodesic and Euclidean distances near the removed strip. All gradient-driven models without injectivity constraints fail to preserve the topological structure of the manifold, stemming from their non-injective encoders. Further analysis (Appendix C.1 and C.2) reveals that gradient-based baselines exhibit strong dependence on the Swiss roll's geometry (curvature and axis length), while graph-based methods vary with sample size. In contrast, BLAE consistently preserves topology across both geometric and population variations.

**dSprites.** The dSprites dataset (Matthey et al., 2017) serves as a benchmark for evaluating disentangled representations. It contains $64 \times 64$ binary images of three geometric primitives (squares, ellipses, and hearts) generated through systematic variation of five factors: shape, color, orientation, scale, and position $(x, y)$. In our experiments, we fix color, scale, and orientation to (white, 1, 0) and select squares and hearts with all possible positions except a cross-shaped region in the center.

We conduct semi-supervised autoencoder training on this shape-partitioned dataset, creating two distinct data clusters by adding a constant value of 1 to all pixels in square images. Figure 4 shows that only BLAE, SPAE, TAE, CAE, and IRAE successfully reconstruct the topological structure

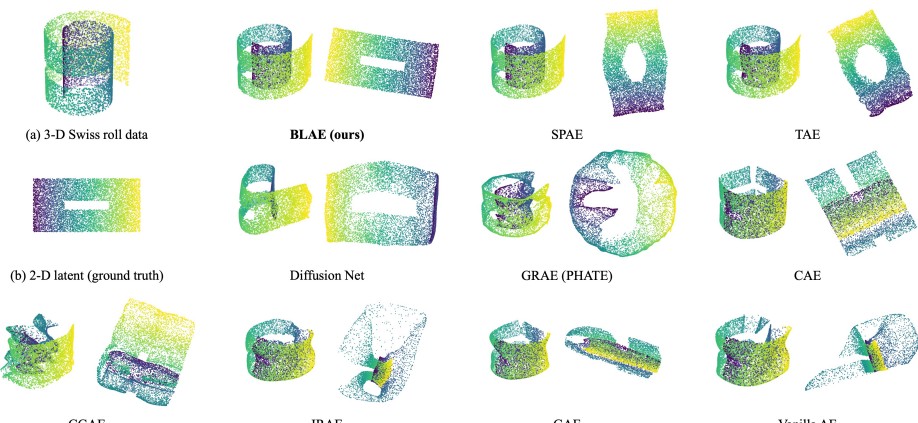

Figure 3: (a) 3-D Swiss roll data. (b) Ground truth: 2-D latent representations to generate a Swiss Roll. *Others:* 3-D reconstruction and 2-D latent representations learned by AE methods.

of both clusters, with BLAE exhibiting the least geometric distortion between the parallel planes representing the two shape classes.

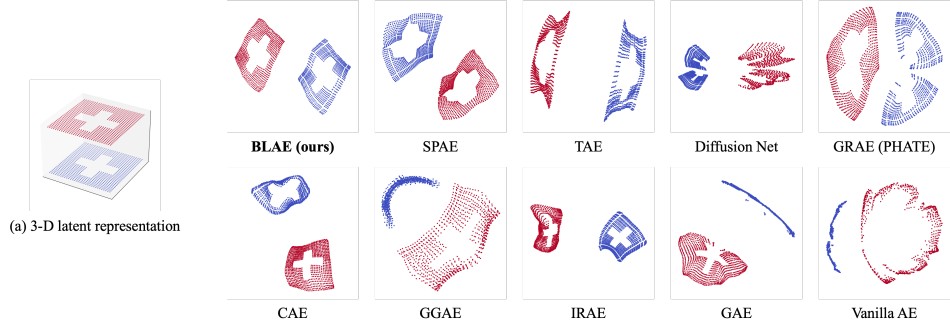

Figure 4: (a) Two parallel planes: 3-D latent representation of square (blue) and heart (red) clusters. *Others:* 2-D latent representations learned by BLAE and other baseline methods.

**MNIST.** To evaluate robustness against distribution shift, we train models using two different sampling distributions from the same underlying manifold. We generate these datasets using distinct rotation strategies on a $28 \times 28$ handwritten digit '3':

1. **Uniform**: Rotate the image in $1°$ increments over the range $(0°, 360°]$, and uniformly sample 25% of these rotations as the training set (Figure 5(b)). The remaining are used as the testing set.
2. **Non-uniform**: Apply $1°$ rotation steps within the ranges $(0°, 30°] \cup (180°, 210°]$, and $10°$ steps within $(30°, 180°] \cup (210°, 360°]$ to obtain the training set, as illustrated in Figure 5(c). The remaining rotation angles constitute the testing set.

Each rotated image is zoomed to five scales ($0.8\times$, $0.9\times$, $1.0\times$, $1.1\times$, $1.2\times$), generating 450 samples distributed across an annular manifold (Figure 5(b) (c)). Figure 5 illustrates the results. Gradient-based models without injectivity constraints struggle to capture the manifold topology, while graph-based methods demonstrate better performance. Notably, although Diffusion Net and TAE preserve topological structure in their latent representations, only BLAE achieves consistent embedding structures—forming concentric circles—across both training distributions, demonstrating its invariance to sampling density variations.

For completeness, Appendix B.4 presents the numerical results for all experiments, Appendix B.5 analyzes the computational complexity, and Appendix C includes extended experiments such as sensitivity analysis, ablation studies, and downstream classification on additional real-world datasets.

## 6 CONCLUSION

This work provides both theoretical and empirical foundations for understanding optimization bottlenecks in autoencoders. We demonstrate that the non-injective nature of standard encoders

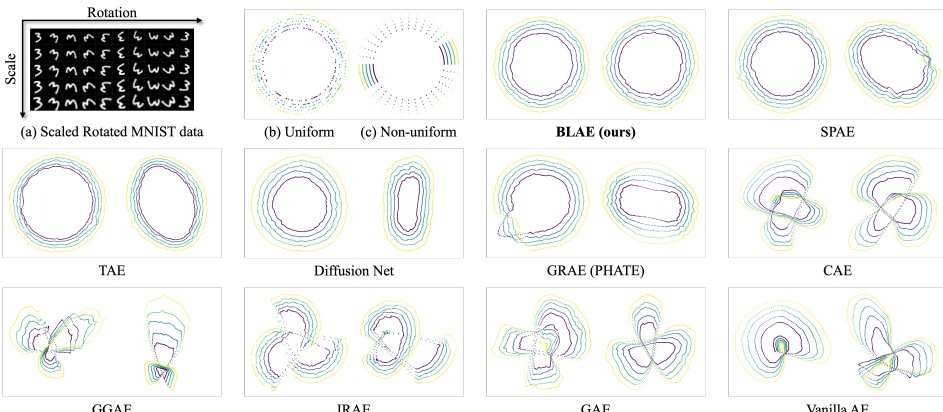

Figure 5: (a) Digit '3' at various scales and rotations. (b) (c) Ground truth: 2D concentric circle latent representation for uniform and non-uniform training sets. *Others:* 2D latent representations learned by BLAE and baseline methods on uniform (left) and non-uniform (right) training sets.

fundamentally induces local minima entrapment, explaining the suboptimal convergence observed in conventional formulations. To address this fundamental limitation, we introduce two key innovations. First, a novel injective regularization framework based on separation criteria that enforces topological consistency between input and latent spaces. Second, a bi-Lipschitz geometric constraint that ensures admissible latent space construction even under aggressive dimensionality reduction. Our Bi-Lipschitz Autoencoder (BLAE) demonstrates state-of-the-art performance in geometric structure preservation across multiple datasets. Importantly, BLAE exhibits significantly enhanced robustness to distribution shifts and low-sample regimes compared to existing approaches, validating our theoretical analysis.

Although this paper primarily compares deterministic autoencoders for manifold learning, it is closely related to deep generative modeling. Many commonly used deep generative models struggle to learn distributions supported on unknown low-dimensional manifolds due to dimensionality mismatch and the phenomenon of manifold overfitting, where likelihood-based models fail to capture the true distribution on the intrinsic manifold Loaiza-Ganem et al. (2024). One common strategy to introduce manifold awareness is to adopt two-step generative architectures, which first perform generative modeling in a low-dimensional latent space and then map samples back to the data space. A variety of existing works Li et al. (2015); Dao et al. (2023); Xiao et al. (2019) provide empirical and theoretical evidence that training generative models in a learned latent representation improves performance compared to training directly in the ambient data space, and it is expected that generative models built on better latent representations achieve even stronger performance. In Appendix C.6.2, we integrate our BLAE framework with a variational autoencoder (VAE), which improves performance compared to standard VAE, further validating that our method is effective for latent generative models as well.

One limitation of training BLAE, common to graph-based methods, lies in the need to precompute the geodesic distance matrix, whose time and space complexities grow quadratically with the number of data points. As a result, on larger-scale datasets (e.g., ImageNet), techniques are needed to mitigate this overhead. For example, one can construct the distance matrix only on a subset of points, and approximate the geodesic distance between any two points using the distance between their nearest neighbors within this subset.

## ACKNOWLEDGMENTS

This work was supported in part by NIH grants U01 AG068057, U01 AG066833, and R01 EB037101. The content is solely the responsibility of the authors and does not necessarily represent the official views of the NIH.

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

# A  THEORETICAL PROOFS

## A.1  PROOF OF THEOREM 1

*Proof.* ($\Leftarrow$) $\forall x \neq y \in \mathcal{M}$, choose $\delta = d_{\mathcal{M}}(x, y)$, there exists $\epsilon > 0$, such that $f$ is $(\delta, \epsilon)$-separated, then

$$\frac{d_{\mathcal{N}}(f(x), f(y))}{d_{\mathcal{M}}(x, y)} > \epsilon. \tag{13}$$

Therefore, $d_{\mathcal{N}}(f(x), f(y)) > \epsilon \cdot d_{\mathcal{M}}(x, y) = \epsilon \cdot \delta > 0$, i.e. $f(x) \neq f(y)$. So $f$ is an injection. Note that the sufficiency does not require any assumption.

($\Rightarrow$) Because $\mathcal{M}$ is compact, $\mathcal{M} \times \mathcal{M}$ is compact as well. For any $\delta > 0$, let

$$C_\delta := \big\{(x, y) \in \mathcal{M} \times \mathcal{M} \mid d_{\mathcal{M}}(x, y) \geq \delta\big\}. \tag{14}$$

The continuity and injectivity of $f$ imply that $d_{\mathcal{N}}(f(x), f(y))/d_{\mathcal{M}}(x, y)$ is continuous and positive on $C_\delta$. Note that $C_\delta$ is a closed subset of $\mathcal{M} \times \mathcal{M}$, and hence also compact. Therefore, there exists $(x_*, y_*) \in C_\delta$ such that

$$\frac{d_{\mathcal{N}}(f(x_*), f(y_*))}{d_{\mathcal{M}}(x_*, y_*)} = \inf_{(x,y) \in C_\delta} \frac{d_{\mathcal{N}}(f(x), f(y))}{d_{\mathcal{M}}(x, y)}. \tag{15}$$

Let $\epsilon = d_{\mathcal{N}}(f(x_*), f(y_*))/(2d_{\mathcal{M}}(x_*, y_*))$, then $f$ is $(\delta, \epsilon)$-separated. $\square$

## A.2  PROOF OF THEOREM 2

*Proof.* Because $R$ has a global minimum, we know that

$$U_{\min} = \arg\min_u R(u) \neq \varnothing. \tag{16}$$

$\forall u_* \in U_{\min}$, we have $R(f(x)) - R(u_*) \geq 0$ for any $x \in \mathcal{M}$. Let $f$ be a minimizer of

$$\min_{f_\Theta \in \mathcal{F}_\Theta} \mathbb{E}_{x \sim \mathbb{P}}\big[R(f_\Theta(x))\big], \tag{17}$$

then

$$0 = \mathbb{E}_{x \sim \mathbb{P}}\big[R(f(x))\big] - R(u_*) = \mathbb{E}_{x \sim \mathbb{P}}\big[R(f_\Theta(x)) - R(u_*)\big] \geq 0. \tag{18}$$

This implies that $R(f(x)) - R(u_*) = 0$ a.s.-$\mathbb{P}$, which means $f(x) \in U_{\min}$ a.s. on $\mathcal{M}$, since $\mathbb{P}$ is strictly positive. Therefore, $f(x) \in U_{\min}$ a.s. on $\mathcal{M}$. On the other hand, if $f(x) \in U_{\min}$ a.s. on $\mathcal{M}$, then since $\mathbb{P}$ is absolutely continuous, we have that $f(x) \in U_{\min}$ a.s.-$\mathbb{P}$. Hence, it is obvious that $f$ is a minimizer of equation 17, so

$$S_{\mathbb{P}} = \{f_\Theta \in \mathcal{F}_\Theta \mid f_\Theta(x) \in U_{\min} \text{ a.s. on } \mathcal{M}\}. \tag{19}$$

Similarly, for another strictly positive and absolutely continuous probability measure $\mathbb{Q}$,

$$S_{\mathbb{Q}} = \{f_\Theta \in \mathcal{F}_\Theta \mid f_\Theta(x) \in U_{\min} \text{ a.s. on } \mathcal{M}\}, \tag{20}$$

i.e. $S_{\mathbb{P}} = S_{\mathbb{Q}}$ which proves the admissibility. $\square$

## A.3  PROOF OF THEOREM 3

Before proving Theorem 3, we first introduce a auxiliary lemma:

**Lemma 1.** Given the assumptions in Theorem 3, we have

$$\sigma_{\max}(J_f^{\mathcal{M}}(x)) = \max_{v \in T_x \mathcal{M} \setminus \{\mathbf{0}\}} \frac{\|J_f(x)v\|}{\|v\|}, \tag{21}$$

$$\sigma_{\min}(J_f^{\mathcal{M}}(x)) = \min_{v \in T_x \mathcal{M} \setminus \{\mathbf{0}\}} \frac{\|J_f(x)v\|}{\|v\|}. \tag{22}$$

*Proof.* Let $k = \dim(T_x\mathcal{M})$, choose $\{v_1, \cdots, v_k\}$ as an orthonormal basis of $T_x\mathcal{M}$. Then we have the following (basis-dependent) representation:

$$J_f^{\mathcal{M}}(x) = \left[ J_f(x)v_1 | \cdots | J_f(x)v_k \right] = J_f(x)V. \tag{23}$$

Note that $J_f^{\mathcal{M}}(x)$ is an $n \times k$ matrix and $k \leq n$ because $f|_M$ is a diffeomorphism, so it has exactly $k$ singular values. We denote them as $\sigma_1 \geq \cdots \geq \sigma_k$. By singular value decomposition

$$J_f^{\mathcal{M}}(x) = P_{n \times n} \Sigma_{n \times k} Q_{k \times k}^\top, \tag{24}$$

where $P, Q$ are orthonormal matrices and $\Sigma = \mathrm{diag}(\sigma_1, \cdots, \sigma_k)$. Then

$$V^\top J_f(x)^\top J_f(x) V = J_f^{\mathcal{M}}(x)^\top J_f^{\mathcal{M}}(x) = Q\Sigma^\top \Sigma Q^\top. \tag{25}$$

So by eigenvalue decomposition,

$$\begin{aligned}
\sigma_{\max}^2(J_f^{\mathcal{M}}(x)) &= \max_{u \in \mathbb{R}^k \setminus \{\mathbf{0}\}} \frac{u^\top \Sigma^\top \Sigma u}{\|u\|^2} \\
&= \max_{w \in \mathbb{R}^k \setminus \{\mathbf{0}\}} \frac{w^\top Q\Sigma^\top \Sigma Q^\top w}{\|w\|^2} \\
&= \max_{w \in \mathbb{R}^k \setminus \{\mathbf{0}\}} \frac{w^\top V^\top J_f(x)^\top J_f(x) V w}{\|w\|^2} \\
&= \max_{v \in T_x\mathcal{M} \setminus \{\mathbf{0}\}} \frac{v^\top J_f(x)^\top J_f(x) v}{\|v\|^2} \\
&= \max_{v \in T_x\mathcal{M} \setminus \{\mathbf{0}\}} \frac{\|J_f(x)v\|^2}{\|v\|^2},
\end{aligned} \tag{26}$$

i.e.

$$\sigma_{\max}(J_f^{\mathcal{M}}(x)) = \max_{v \in T_x\mathcal{M} \setminus \{\mathbf{0}\}} \frac{\|J_f(x)v\|}{\|v\|}. \tag{27}$$

Similarly,

$$\sigma_{\min}(J_f^{\mathcal{M}}(x)) = \min_{v \in T_x\mathcal{M} \setminus \{\mathbf{0}\}} \frac{\|J_f(x)v\|}{\|v\|}. \tag{28}$$

$\square$

*Proof of Theorem 3.* ($\Rightarrow$) Suppose $f$ is $\kappa$-bi-Lipschitz, $\forall x \in \mathrm{int}\, \mathcal{M}$, consider a unit vector $v \in T_x\mathcal{M}$. Let $\gamma : (-\varepsilon, \varepsilon) \to \mathcal{M}$ be a smooth curve with $\gamma(0) = x$ and $\gamma'(0) = v$. By the chain rule:

$$(f \circ \gamma)'(0) = J_f(x)v. \tag{29}$$

For $|t| < \varepsilon$, the bi-Lipschitz condition implies that

$$\frac{1}{\kappa} \cdot d_{\mathcal{M}}(\gamma(t), x) \leq d_{\mathcal{N}}(f(\gamma(t)), f(x)) \leq \kappa \cdot d_{\mathcal{M}}(\gamma(t), x). \tag{30}$$

Through dividing by $|t|$ and taking $t \to 0$, we obtain:

$$\frac{1}{\kappa}\|v\| \leq \|J_f(x)v\| \leq \kappa\|v\|, \quad \forall v \in T_x\mathcal{M}. \tag{31}$$

For $v \neq \mathbf{0}$, dividing equation 31 by $\|v\|$ and taking maximum (minimum) give:

$$\frac{1}{\kappa} \leq \min_{v \in T_x\mathcal{M} \setminus \{\mathbf{0}\}} \frac{\|J_f(x)v\|}{\|v\|} \leq \max_{v \in T_x\mathcal{M} \setminus \{\mathbf{0}\}} \frac{\|J_f(x)v\|}{\|v\|} \leq \kappa. \tag{32}$$

By lemma 1,

$$\frac{1}{\kappa} \leq \sigma_{\min}(J_f(x)) \leq \sigma_{\max}(J_f(x)) \leq \kappa, \quad \forall x \in \mathrm{int}\, \mathcal{M}. \tag{33}$$

By Weyl's inequality, equation 33 holds for all $x \in \overline{\mathrm{int}\, \mathcal{M}} = \mathcal{M}$.

($\Leftarrow$) Let $x, y \in \mathcal{M}$ and suppose $\gamma : [0, 1] \to \mathcal{M}$ is a smooth path from $x$ to $y$ with $\|\gamma'\| > 0$ (such path must exist because $\mathcal{M}$ is path-connected). Then the image of $\gamma$ is a path from $f(x)$ to $f(y)$.

$$
\begin{aligned}
\text{Length}(f|_{\mathcal{M}} \circ \gamma) &= \int_0^1 \|(f|_{\mathcal{M}} \circ \gamma)'(s)\| ds \\
&= \int_0^1 \|J_f(\gamma(s))\gamma'(s)\| ds \quad (\gamma'(s) \in T_s \mathcal{M}) \\
&= \int_0^1 \frac{\|J_f(\gamma(s))\gamma'(s)\|}{\|\gamma'(s)\|} \cdot \|\gamma'(s)\| ds \\
&\leq \int_0^1 \max_{v \in T_s \mathcal{M} \setminus \{\mathbf{0}\}} \frac{\|J_f(\gamma(s))v\|}{\|v\|} \cdot \|\gamma'(s)\| ds \\
&\leq \int_0^1 \kappa \cdot \|\gamma'(s)\| ds \\
&= \kappa \cdot \text{Length}(\gamma).
\end{aligned}
\tag{34}
$$

Because $f|_{\mathcal{M}}$ is a diffeomorphism, $f|_{\mathcal{M}}^{-1}$ exists and is smooth, hence for any smooth path $\beta : [0, 1] \to f(\mathcal{M})$ from $f(x)$ to $f(y)$, $f^{-1}|_{\mathcal{M}} \circ \beta$ must be a smooth path from $x$ to $y$. So, by the definition of geodesic distance,

$$
\begin{aligned}
d_{\mathcal{N}}(f(x), f(y)) &= \inf_\beta \text{Length}(\beta) \\
&= \inf_\beta \text{Length}(f|_{\mathcal{M}} \circ f|_{\mathcal{M}}^{-1} \circ \beta) \\
&= \inf_\gamma \text{Length}(f|_{\mathcal{M}} \circ \gamma) \\
&\leq \kappa \cdot \inf_\gamma \text{Length}(\gamma) \\
&= \kappa \cdot d_{\mathcal{M}}(x, y).
\end{aligned}
\tag{35}
$$

Similarly, we can prove that $d_{\mathcal{N}}(f(x), f(y)) \geq \frac{1}{\kappa} \cdot d_{\mathcal{M}}(x, y)$ which completes the proof. $\qquad\square$

### A.4 PROOF OF THEOREM 4

Although this theorem can be proved in just a few lines using Proposition C.29 and the Inverse Function Theorem in (Lee, 2003), for the sake of completeness we present here a more detailed proof.

*Proof.* Suppose $f$ embeds $\mathcal{M}$ into $R^n$. The Whitney's embedding theorem guarantees that such an embedding exists and $n \leq 2k$. On the other hand, the embedding dimension obviously cannot be less than $k$. It remains to show that $f$ is bi-Lipschitz. Since $f$ is an embedding, we know that the linear mapping $J_f(x) : T_x \mathcal{M} \to T_{f(x)} f(\mathcal{M})$ is injective at every point $x \in \mathcal{M}$. Thereby, for any unit vector $v \in T_x \mathcal{M}$,

$$
\|J_f(x)v\| > 0,
\tag{36}
$$

Because $V := \{v : \big| v \in T_x \mathcal{M}, \|v\| = 1\}$ is compact, there must exist a $v_* \in V$ such that

$$
\sigma_{\min}(J_f^{\mathcal{M}}(x)) = \min_{v \in T_x \mathcal{M} \setminus \{\mathbf{0}\}} \frac{\|J_f(x)v\|}{\|v\|} = \min_{v \in V} \|J_f(x)v\| = \|J_f(x)v_*\| > 0.
\tag{37}
$$

By Weyl's inequality (which implies the continuity of singular values of $J_f$), there exists a neighborhood $U_x$ of $x$, such that $\forall y \in U_x$,

$$
\sigma_{\min}(J_f^{\mathcal{M}}(y)) \geq \frac{1}{2} \sigma_{\min}(J_f^{\mathcal{M}}(x)),
\tag{38}
$$

$$
\sigma_{\max}(J_f^{\mathcal{M}}(y)) \leq 2\sigma_{\max}(J_f^{\mathcal{M}}(x)).
\tag{39}
$$

Choose $\kappa(x) = \max\{2\sigma_{\max}(J_f^{\mathcal{M}}(x)), 2/\sigma_{\min}(J_f^{\mathcal{M}}(x))\}$, it is easy to see that $\kappa(x) \geq 1$, we have:

$$
\frac{1}{\kappa(x)} \leq \sigma_{\min}(J_f^{\mathcal{M}}(y)) \leq \sigma_{\max}(J_f^{\mathcal{M}}(y)) \leq \kappa(x), \quad \forall y \in U_x.
\tag{40}
$$

Notice that $\{U_x | x \in \mathcal{M}\}$ is a open cover of $\mathcal{M}$, since $\mathcal{M}$ is compact, we can find a finite sub-cover $\{U_{x_1}, \cdots, U_{x_N}\}$. Choose $\kappa = \max\{\kappa(x_1), \cdots, \kappa(x_N)\}$, then,

$$\frac{1}{\kappa} \leq \sigma_{\min}(J_f^{\mathcal{M}}(y)) \leq \sigma_{\max}(J_f^{\mathcal{M}}(y)) \leq \kappa, \quad \forall y \in \mathcal{M}. \tag{41}$$

By Theorem 3, $f$ is $\kappa$-bi-Lipschitz. $\qquad \square$

## B  EXPERIMENTAL DETAILS

### B.1  HYPERPARAMETER

We implemented dataset-specific autoencoders tailored to the structural characteristics of each dataset. For the Swiss Roll and ssREAD datasets, we used a fully connected autoencoder with two hidden layers of 256 units and ELU activations in both the encoder and decoder. The encoder projected the input—either 3D coordinates (Swiss Roll) or 50-dimensional PCA-reduced features (ssREAD)—into a 2-dimensional latent space, which the decoder then used to reconstruct the original input. For Swiss Roll, models were trained for 3000 epochs using the Adam optimizer with weight decay $1 \times 10^{-5}$ and an initial learning rate of $2 \times 10^{-3}$, reduced by a factor of 0.1 every 1000 epochs. For ssREAD, models were trained for 1500 epochs with Adam, weight decay $1 \times 10^{-5}$, and an initial learning rate of $1 \times 10^{-3}$, reduced by a factor of 0.1 every 500 epochs.

For the dSprites dataset, which is detailed in Appendix C.6, we adopted a convolutional autoencoder based on the ConvNet64 and DeConvNet64 architectures. The encoder consisted of five convolutional layers with increasing channel widths—from $n_h$ to $4 \times n_h$—followed by a $1 \times 1$ convolution that mapped the feature maps to a 2D latent representation. The decoder mirrored this structure using transposed convolutions to reconstruct the original $64 \times 64$ binary image. Training was conducted for 1000 epochs using Adam with weight decay $1 \times 10^{-5}$ and an initial learning rate of $1 \times 10^{-3}$.

For the MNIST dataset, we employed a convolutional autoencoder optimized for $28 \times 28$ grayscale images. The encoder consisted of two convolutional layers with ReLU activations and max pooling, followed by fully connected layers that compressed the input into a 2D latent vector. The decoder performed the inverse, using fully connected layers and transposed convolutions to reconstruct the image. Models were trained for 3000 epochs with Adam, weight decay $1 \times 10^{-5}$, and an initial learning rate of $1 \times 10^{-2}$, reduced by a factor of 0.5 every 300 epochs.

All models were designed to produce a 2-dimensional latent space to facilitate direct visualization and consistent comparison.

Table 2: Hyperparameter settings for baselines across all evaluated datasets.

| Dataset | Parameter | SPAE | TAE | DN | GRAE | CAE | GGAE | IRAE | GAE |
|---|---|---|---|---|---|---|---|---|---|
| Swiss Roll | $\lambda$ | 2 | 5 | 100 | 100 | 0.1 | 1 | 0.01 | 0.01 |
| | $\eta$ | / | / | 0.001 | / | / | / | 0 | / |
| | n_neighbor | / | / | 10 | 5 | / | 10 | / | / |
| | bandwidth | / | / | / | / | / | 2 | / | / |
| dSprites | $\lambda$ | 10 | 0.01 | 0.001 | 0.1 | 0.1 | 0.1 | 0.1 | 0.01 |
| | $\eta$ | / | / | 0.1 | / | / | / | 0 | / |
| | n_neighbor | / | / | 1000 | 6 | / | 5 | / | / |
| | bandwidth | / | / | / | / | / | 5 | / | / |
| MNIST | $\lambda$ | 5 | 10 | 0.01 | 0.01 | 0.1 | 0.1 | 0.1 | 0.01 |
| | $\eta$ | / | / | 0.1 | / | / | / | 0 | / |
| | n_neighbor | / | / | 50 | 5 | / | 15 | / | / |
| | bandwidth | / | / | / | / | / | 0.01 | / | / |
| ssREAD | $\lambda$ | 10 | 100 | 0.01 | 0.01 | 1 | 0.01 | 1 | 1 |
| | $\eta$ | / | / | 0.1 | / | / | / | 0.2 | / |
| | n_neighbor | / | / | 5 | 5 | / | 10 | / | / |
| | bandwidth | / | / | / | / | / | 0.01 | / | / |

The hyperparameters used for the baseline models on each dataset are listed in Table 2. These were selected via grid search: $\lambda \in \{0.01, 0.1, 1, 2, 5, 10, 100\}$, $\eta \in \{0.001, 0.01, 0.1, 0.2, 0.5, 1\}$, number of neighbors $\in \{5, 6, 7, 8, 9, 10, 15, 50, 100, 1000\}$, and GGAE bandwidth $\in \{0.01, 0.1, 1, 2, 5, 10\}$.

In our method, $\kappa$ is treated as a tunable hyperparameter. A simple approach is to apply a standard grid search, but a more efficient alternative is to use a binary search over a given interval (e.g., $[1, 5]$). Starting from the midpoint, if the trained model produces a nonzero Bi-Lipschitz loss (above a tolerance such as $10^{-4}$), this indicates that the current $\kappa$ is too small and should be increased; if the loss is effectively zero, the condition is satisfied and $\kappa$ can be accepted or further reduced. In practice, the guiding principle is to select the smallest $\kappa$ for which the Bi-Lipschitz loss remains below the threshold. Alternatively, if one wishes to enforce a specific distortion bound $K$ between the latent representations and the input space, $\kappa$ can be directly set to $K$, thereby ensuring the constraint while maintaining robustness to distributional shift.

The hyperparameters used for our model (BLAE) are provided in Table 3.

Table 3: Hyperparameter settings for BLAE across all evaluated datasets.

| Datasets | Swiss Roll | dSprites | MNIST | ssREAD |
|---|---|---|---|---|
| $\lambda_{\text{reg}}$ | 1 | 2 | 30 | 2 |
| $\lambda_{\text{bi-Lip}}$ | 0.3 | 0.1 | 0.1 | 0.1 |
| $\kappa$ | 1 | 1.1 | 2 | 1.2 |
| $\epsilon$ | 0.3 | 0.3 | 0.6 | 0.6 |

## B.2 EVALUATION METRICS

We evaluate the performance of each model using three metrics: mean squared error (MSE), $k$-NN recall (Sainburg et al., 2021; Kobak et al., 2019), and $KL_\sigma$ divergence (Chazal et al., 2011), where $\sigma \in 0.01, 0.1, 1$. While these metrics are traditionally defined using Euclidean distance, we adapt them to better capture the underlying manifold structure. Specifically, we compute distances using geodesic metrics: $d_\mathcal{M}$ on the data manifold and $d_\mathcal{N}$ in the latent space. These geodesic distances are approximated by first constructing a neighborhood graph and then computing the shortest paths between node pairs.

### B.2.1 $k$-NN RECALL

The $k$-NN recall metric quantifies how well the local neighborhood structure is preserved in the latent space. It measures the proportion of $k$-nearest neighbors on the data manifold that remain among the $k$-nearest neighbors in the latent space. For the Swiss Roll dataset, we report the average $k$-NN recall over $k \in \{5, 10, \dots, 50\}$. For MNIST and dSprites, we use $k \in \{2, 4, \dots, 10\}$.

### B.2.2 $KL_\sigma$ DIVERGENCE

The $KL_\sigma$ metric computes the Kullback-Leibler divergence between the normalized density estimates on the data manifold and in the latent space. Let $X = \{x_1, \dots, x_N\}$ denote the original data and $Z = \{z_1, \dots, z_N\}$ their corresponding latent representations. The density at each point is defined by:

$$p_{X,\sigma}(x_i) = \frac{f_{X,\sigma}(x_i)}{\sum_j f_{X,\sigma}(x_j)}, \qquad p_{Z,\sigma}(z_i) = \frac{f_{Z,\sigma}(z_i)}{\sum_j f_{Z,\sigma}(z_j)}, \tag{42}$$

where the unnormalized densities are computed as:

$$\begin{aligned} f_{X,\sigma}(x_i) &= \sum_j \exp\left(-\frac{1}{\sigma}\left(\frac{d_\mathcal{M}(x_i, x_j)}{\max_{i,j} d_\mathcal{M}(x_i, x_j)}\right)^2\right), \\ f_{Z,\sigma}(z_i) &= \sum_j \exp\left(-\frac{1}{\sigma}\left(\frac{d_\mathcal{N}(z_i, z_j)}{\max_{i,j} d_\mathcal{N}(z_i, z_j)}\right)^2\right). \end{aligned} \tag{43}$$

The final metric is given by $KL_\sigma = D_{\text{KL}}(p_{X,\sigma} \| p_{Z,\sigma})$. Smaller values of $\sigma$ emphasize local structure, while larger values reflect global geometry.

### B.3 GENERATION OF SWISS ROLL DATA

We use the logarithmic spiral $r = e^{b\theta}$ $(b \neq 0)$ to construct the Swiss Roll dataset. The arc length of the spiral from angle $\theta_1$ to $\theta_2$ is given by:

$$
\begin{aligned}
s(\theta_2) - s(\theta_1) &= \int_{\theta_1}^{\theta_2} 1 \, \mathrm{d}s \\
&= \int_{\theta_1}^{\theta_2} \sqrt{r^2(\theta) + r'^2(\theta)} \mathrm{d}\theta \\
&= \int_{\theta_1}^{\theta_2} e^{b\theta} \sqrt{1 + b^2} \mathrm{d}\theta \\
&= \frac{\sqrt{1 + b^2}}{b} (e^{b\theta_2} - e^{b\theta_1}).
\end{aligned}
\tag{44}
$$

Fixing the starting point at $\theta_1 = 0$ and allowing the negative arc length to be negative, we obtain the arc length as a function of $\theta$:

$$
s(\theta) = \frac{\sqrt{1 + b^2}}{b} (e^{b\theta} - 1),
\tag{45}
$$

which leads to the inverse function:

$$
\theta(s) = \frac{1}{b} \log\left(\frac{bs}{\sqrt{1 + b^2}} + 1\right).
\tag{46}
$$

This yields an isometric parameterization of the logarithmic spiral over the interval $(-\frac{\sqrt{1+b^2}}{b}, +\infty)$:

$$
r(s) = e^{b\theta(s)}.
\tag{47}
$$

To generate the Swiss Roll, we first uniformly sample points $(s, z) \in [-2, 10] \times [0, 6]$, then remove points within the rectangular strip $[1.5, 6.5] \times [2.5, 3.5]$. The remaining points are embedded isometrically into $\mathbb{R}^3$ via:

$$
(s, z) \rightarrow (e^{b\theta(s)} \cos(\theta(s)), e^{b\theta(s)} \sin(\theta(s)), z).
\tag{48}
$$

In Section 5, we set $b = 0.1$.

### B.4 QUANTITATIVE RESULTS

In this section, we present quantitative results, including mean squared error (MSE), $k$-nearest-neighbor ($k$-NN) recall, and KL divergence ($KL_\sigma$) under multiple bandwidths $\sigma \in \{0.01, 0.1, 1\}$, for the Swiss Roll (Table 4), dSprites (Table 5), and MNIST datasets(Table 6 and Table 7). For dSprites, evaluation is performed specifically on the withheld cross-shaped regions to assess model generalization to out-of-distribution samples. For each cluster, we compute MSE, $k$-NN recall, and $KL_\sigma$, and report the results as the mean $\pm$ standard deviation over five independent runs, with the final score for each model given by the average across clusters. The best performance for each metric is highlighted in bold.

The Swiss Roll results show that BLAE attains the lowest reconstruction error and KL divergence across all bandwidths, together with the highest $k$-NN recall, indicating both accurate geometry recovery and strong neighborhood preservation. On dSprites, BLAE also achieves the best scores across metrics, with particularly large margins on $KL_\sigma$, demonstrating generalization to out-of-distribution regions. For uniform MNIST, BLAE remains competitive on all measures, leading in MSE, $k$-NN, and KL at three bandwidths. On the more challenging non-uniform MNIST, although some baselines approach similar $k$-NN performance, BLAE secures the best results on three of the five metrics, and the embeddings in Figure 5 show markedly greater robustness to distributional shift.

### B.5 COMPUTATIONAL COMPLEXITY

BLAE, along with other graph-based autoencoders, begins by constructing a neighborhood graph and computing pairwise shortest paths to approximate geodesic distances. This preprocessing step has

Table 4: Evaluation metrics (mean ± standard deviation over 5 runs) on the Swiss Roll dataset. For MSE and KL metrics, lower values are better; for $k$-NN, higher values are better.

| Model | $k$-NN($\uparrow$) | $KL_{0.01}$($\downarrow$) | $KL_{0.1}$($\downarrow$) | $KL_1$($\downarrow$) | MSE($\downarrow$) |
|---|---|---|---|---|---|
| BLAE | **9.61e-01 ± 3.59e-03** | **1.28e-04 ± 2.95e-05** | **1.78e-05 ± 5.98e-06** | **1.39e-06 ± 1.02e-06** | **9.36e-05 ± 1.92e-05** |
| SPAE | 8.95e-01 ± 2.90e-02 | 7.95e-03 ± 1.13e-02 | 5.15e-03 ± 8.95e-03 | 4.73e-04 ± 8.81e-04 | 2.10e-03 ± 3.25e-03 |
| TAE | 8.14e-01 ± 1.54e-02 | 4.28e-03 ± 7.20e-04 | 1.57e-03 ± 6.38e-04 | 5.64e-05 ± 1.72e-05 | 6.94e-03 ± 5.07e-03 |
| DN | 7.22e-01 ± 1.58e-02 | 5.26e-02 ± 7.95e-03 | 1.32e-02 ± 7.50e-03 | 7.43e-04 ± 7.03e-04 | 1.02e-01 ± 1.73e-02 |
| GRAE | 5.70e-01 ± 3.03e-02 | 4.09e-02 ± 1.26e-02 | 9.94e-03 ± 4.36e-03 | 8.34e-04 ± 4.11e-04 | 7.72e-02 ± 2.16e-02 |
| CAE | 6.14e-01 ± 1.71e-02 | 6.11e-02 ± 1.66e-02 | 4.10e-02 ± 7.76e-03 | 2.31e-03 ± 4.27e-04 | 5.37e-02 ± 7.41e-03 |
| GGAE | 6.61e-01 ± 1.22e-01 | 4.04e-02 ± 2.52e-02 | 1.80e-02 ± 1.09e-02 | 1.18e-03 ± 7.70e-04 | 7.28e-02 ± 4.96e-02 |
| IRAE | 5.88e-01 ± 1.45e-02 | 2.55e-01 ± 2.15e-02 | 6.75e-02 ± 8.34e-03 | 2.31e-03 ± 4.82e-04 | 4.06e-02 ± 2.08e-03 |
| GAE | 5.03e-01 ± 3.21e-02 | 6.96e-02 ± 3.56e-02 | 2.73e-02 ± 1.28e-02 | 1.92e-03 ± 4.27e-04 | 4.80e-02 ± 6.22e-04 |
| Vanilla AE | 5.18e-01 ± 2.90e-02 | 2.00e-01 ± 9.14e-02 | 5.41e-02 ± 2.07e-02 | 2.05e-03 ± 6.25e-04 | 4.14e-02 ± 3.09e-02 |

Table 5: Evaluation metrics (mean ± standard deviation over 5 runs) on the dSprites dataset. For MSE and KL metrics, lower values are better; for $k$-NN, higher values are better.

| Model | $k$-NN($\uparrow$) | $KL_{0.01}$($\downarrow$) | $KL_{0.1}$($\downarrow$) | $KL_1$($\downarrow$) | MSE($\downarrow$) |
|---|---|---|---|---|---|
| BLAE | **7.39e-01 ± 5.17e-03** | **6.42e-03 ± 9.52e-04** | **3.98e-03 ± 1.84e-04** | **3.79e-05 ± 1.74e-06** | **1.69e-02 ± 1.46e-03** |
| SPAE | 7.24e-01 ± 4.60e-03 | 2.95e-02 ± 4.67e-04 | 9.11e-03 ± 5.36e-05 | 7.83e-05 ± 4.92e-07 | 1.72e-02 ± 1.02e-03 |
| TAE | 5.58e-01 ± 5.09e-02 | 3.15e-02 ± 2.44e-02 | 1.46e-02 ± 5.37e-03 | 1.25e-03 ± 2.23e-03 | 2.84e-02 ± 4.65e-03 |
| DN | 4.81e-01 ± 4.59e-02 | 8.41e-02 ± 4.45e-02 | 7.32e-02 ± 5.16e-02 | 6.38e-03 ± 5.67e-03 | 3.47e-02 ± 3.55e-02 |
| GRAE | 5.23e-01 ± 1.99e-02 | 2.94e-02 ± 4.97e-04 | 9.10e-03 ± 5.59e-05 | 7.84e-05 ± 4.72e-07 | 3.06e-02 ± 8.94e-03 |
| CAE | 6.79e-01 ± 6.31e-02 | 3.99e-02 ± 2.38e-02 | 2.05e-02 ± 2.54e-02 | 1.79e-03 ± 3.83e-03 | 2.59e-02 ± 8.76e-03 |
| GGAE | 4.36e-01 ± 1.10e-01 | 1.83e-01 ± 5.64e-02 | 2.95e-01 ± 1.33e-01 | 2.72e-03 ± 1.57e-03 | 5.19e-02 ± 6.98e-03 |
| IRAE | 4.89e-01 ± 2.25e-02 | 1.11e-01 ± 3.51e-02 | 3.13e-02 ± 1.98e-02 | 3.32e-03 ± 1.84e-03 | 5.57e-02 ± 3.09e-03 |
| GAE | 4.99e-01 ± 1.12e-01 | 3.73e-02 ± 2.35e-02 | 1.79e-02 ± 1.84e-02 | 1.68e-03 ± 3.57e-03 | 4.36e-02 ± 8.35e-03 |
| Vanilla AE | 4.89e-01 ± 6.01e-02 | 1.21e-01 ± 7.22e-02 | 5.23e-02 ± 3.44e-02 | 4.71e-03 ± 5.16e-03 | 4.79e-02 ± 4.05e-03 |

Table 6: Evaluation metrics (mean ± standard deviation over 5 runs) on the (Uniform) MNIST dataset. For MSE and KL metrics, lower values are better; for $k$-NN, higher values are better.

| Model | $k$-NN($\uparrow$) | $KL_{0.01}$($\downarrow$) | $KL_{0.1}$($\downarrow$) | $KL_1$($\downarrow$) | MSE($\downarrow$) |
|---|---|---|---|---|---|
| BLAE | **9.03e-01 ± 9.38e-03** | **4.79e-02 ± 5.32e-03** | **4.01e-02 ± 1.39e-02** | **1.22e-02 ± 6.28e-03** | **2.92e-03 ± 1.53e-03** |
| SPAE | 8.59e-01 ± 3.10e-02 | 7.50e-02 ± 1.71e-02 | 6.35e-02 ± 1.42e-02 | 1.58e-02 ± 7.63e-03 | 6.91e-03 ± 2.29e-03 |
| TAE | 8.29e-01 ± 2.68e-02 | 5.17e-02 ± 9.51e-03 | 6.69e-02 ± 1.99e-02 | 1.64e-02 ± 8.08e-03 | 4.93e-03 ± 6.78e-04 |
| DN | 8.64e-01 ± 3.17e-02 | 1.33e-01 ± 4.48e-02 | 9.15e-02 ± 3.60e-02 | 1.49e-02 ± 6.72e-03 | 3.43e-03 ± 9.97e-04 |
| GRAE | 8.70e-01 ± 5.67e-02 | 1.11e-01 ± 5.63e-02 | 6.87e-02 ± 2.43e-02 | 1.41e-02 ± 7.06e-03 | 3.51e-03 ± 1.32e-03 |
| CAE | 7.69e-01 ± 3.23e-02 | 2.58e-01 ± 3.32e-02 | 1.62e-01 ± 4.66e-02 | 1.93e-02 ± 8.74e-03 | 5.54e-03 ± 8.10e-04 |
| GGAE | 8.11e-01 ± 4.09e-02 | 4.95e-01 ± 1.54e-01 | 1.95e-01 ± 4.04e-02 | 1.96e-02 ± 8.65e-03 | 4.33e-03 ± 1.69e-03 |
| IRAE | 7.82e-01 ± 4.04e-02 | 1.04e-01 ± 3.58e-02 | 9.41e-02 ± 2.13e-02 | 1.65e-02 ± 8.70e-03 | 6.17e-03 ± 4.27e-04 |
| GAE | 7.03e-01 ± 3.81e-02 | 1.31e-01 ± 4.19e-02 | 1.31e-01 ± 4.31e-02 | 1.96e-02 ± 9.24e-03 | 6.59e-03 ± 1.27e-03 |
| Vanilla AE | 7.69e-01 ± 6.32e-02 | 4.27e-01 ± 1.65e-01 | 1.76e-01 ± 4.96e-02 | 1.85e-02 ± 8.32e-03 | 6.58e-03 ± 4.79e-03 |

Table 7: Evaluation metrics (mean ± standard deviation over 5 runs) on the (non-uniform) MNIST dataset. For MSE and KL metrics, lower values are better; for $k$-NN, higher values are better.

| Model | $k$-NN($\uparrow$) | $KL_{0.01}$($\downarrow$) | $KL_{0.1}$($\downarrow$) | $KL_1$($\downarrow$) | MSE($\downarrow$) |
|---|---|---|---|---|---|
| BLAE | 8.65e-01 ± 4.50e-02 | **2.89e-02 ± 1.80e-02** | **1.66e-02 ± 2.60e-03** | **1.19e-03 ± 7.91e-05** | 3.88e-03 ± 7.87e-04 |
| SPAE | 8.22e-01 ± 1.05e-01 | 5.59e-02 ± 3.13e-02 | 6.11e-02 ± 4.39e-02 | 5.44e-03 ± 5.09e-03 | 7.78e-03 ± 2.90e-03 |
| TAE | 8.77e-01 ± 1.22e-02 | 3.35e-02 ± 8.59e-03 | 2.23e-02 ± 1.01e-03 | 1.32e-03 ± 1.66e-05 | 5.61e-03 ± 1.88e-03 |
| DN | 8.81e-01 ± 6.48e-02 | 7.19e-02 ± 4.58e-02 | 4.10e-02 ± 1.74e-02 | 4.23e-03 ± 2.03e-03 | 5.48e-03 ± 1.65e-03 |
| GRAE | **9.14e-01 ± 2.22e-02** | 1.06e-01 ± 1.57e-02 | 4.84e-02 ± 1.72e-02 | 5.23e-03 ± 1.30e-03 | **3.08e-03 ± 8.84e-04** |
| CAE | 7.58e-01 ± 4.05e-02 | 1.48e-01 ± 1.95e-02 | 1.12e-01 ± 5.71e-03 | 4.55e-03 ± 2.44e-04 | 8.50e-03 ± 1.33e-03 |
| GGAE | 7.15e-01 ± 4.59e-02 | 2.63e-01 ± 9.76e-02 | 1.52e-01 ± 1.06e-01 | 8.92e-03 ± 1.03e-02 | 9.75e-03 ± 2.48e-03 |
| IRAE | 7.36e-01 ± 6.63e-02 | 7.25e-02 ± 1.85e-02 | 7.93e-02 ± 2.35e-02 | 5.17e-03 ± 3.55e-03 | 8.75e-03 ± 2.46e-03 |
| GAE | 6.82e-01 ± 1.94e-01 | 1.48e-01 ± 6.16e-02 | 9.22e-02 ± 2.45e-02 | 4.13e-03 ± 8.02e-04 | 1.45e-02 ± 1.90e-02 |
| Vanilla AE | 7.49e-01 ± 4.75e-02 | 2.76e-01 ± 1.07e-01 | 1.48e-01 ± 3.72e-02 | 7.74e-03 ± 3.39e-03 | 9.70e-03 ± 1.75e-03 |

a time complexity of $\mathcal{O}(n^2 \log n)$, but it is performed only once and thus does not impact training efficiency. During training, each mini-batch only requires slicing the precomputed geodesic distance matrix to extract the relevant submatrix. For instance, in the Swiss Roll experiment, computing the full geodesic distance matrix took only 0.3 seconds.

Although BLAE integrates both graph-based (injective) and gradient-based (bi-Lipschitz) regularization mechanisms, its computational complexity does not significantly exceed that of employing either regularization approach in isolation. This efficiency stems from the fact that only a small subset of samples activate the regularization terms during practical training. Specifically, for data pair $(x, x')$ satisfying $d_{\mathcal{N}}(x, x')/d_{\mathcal{M}}(x, x') \in (\epsilon, 1)$ or data point $x$ where $1/\kappa < \sigma_{\min}(J_{\mathcal{D}_\phi}(x)) \leq \sigma_{\max}(J_{\mathcal{D}_\phi}(x)) < \kappa$, the corresponding gradients of regularization vanish identically.

Table 8: Runtime for training BLAE and other baseline models. All experiments were conducted on a Mac Mini equipped with an Apple M4 chip (16GB RAM).

| Model | BLAE | SPAE | TAE | DN | GRAE | CAE | GGAE | IRAE | GAE | Vanilla AE |
|---|---|---|---|---|---|---|---|---|---|---|
| Runtime (s) | 170.9 | 162.1 | 601.3 | 204.4 | 147.3 | 199.6 | 415.8 | 189.3 | 207.6 | 136.2 |

Consequently, these inactive components are naturally excluded from backpropagation computations. To benchmark, we measured the runtime for training the Swiss Roll dataset in 3000 training steps with 1500 samples divided into three batches (batch size=500). Notably, considering the robustness of gradient-based regularization under limited sample sizes, we implemented a partial sampling strategy where only 10% of data points within each training batch are utilized for computing the Jacobian matrix and the corresponding regularization. As shown in Table 8, BLAE exhibits comparable time complexity to other baselines.

## C  EXTENTED EXPERIMENTS

### C.1  SENSITIVITY ANALYSIS OF $b$ AND LENGTH

In this section, we conduct a sensitivity analysis on the Swiss Roll dataset with respect to two key parameters: the spiral factor $b$ and the manifold length, both of which influence the geometric complexity of the data manifold.

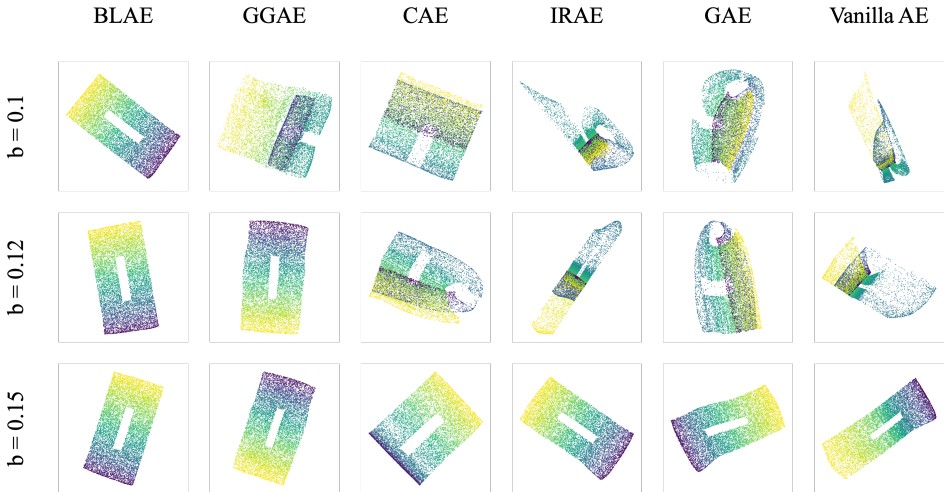

Figure 6: 2-D latent representations of Swiss Roll data learned by BLAE and gradient-based baselines trained with different values of $b$ (0.1, 0.12, 0.15) on the Swiss Roll data. All models were trained using 1500 sample points, and all figures were plotted using the entire 10,000 sample points.

In logarithmic spirals, the curvature is inversely related to $|b|$: smaller values of $|b|$ correspond to tighter coiling of the spiral. During experiments, we observed that gradient-based autoencoders, such

as TAE, SPAE, and GGAE, are more prone to converge to suboptimal local minima as $|b|$ decreases. To assess this behavior systematically, we varied b while keeping all other parameters fixed. As shown in Figure 6, when $b = 0.1$, the baseline models struggle to fully unfold the manifold structure. Interestingly, at $b = 0.12$, GGAE shows earlier signs of escaping non-injective regimes, likely due to its hybrid design that combines gradient-based regularization with structural information from a graph. Full unfolding of the spiral is achieved by all models once $b$ increases to 0.15.

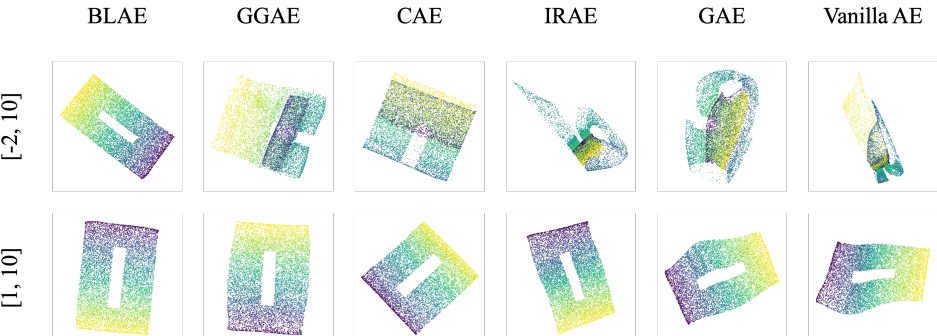

Figure 7: 2-D latent representations learned of Swiss Roll data by BLAE and graph-based baselines trained with different lengths ([-2, 10], [1, 10]) on the Swiss Roll data. All models were trained using 1500 sample points, and all figures were plotted using the entire 10,000 sample points.

The manifold length is another important factor influencing unfolding quality. Empirical evidence suggests that gradient-based models tend to perform better on shorter Swiss Roll configurations. To evaluate this, we fixed $b = 0.1$ and generated two versions of the Swiss Roll: (1) a longer manifold defined over $[-2, 10] \times [0, 6] \setminus [1.5, 6.5] \times [2.5, 3.5]$, and (2) a shorter manifold over $[1, 10] \times [0, 6] \setminus [3.5, 7.5] \times [2.5, 3.5]$. As shown in Figure 7, the shorter configuration leads to more effective unfolding for most models, particularly those graph-based baseline methods.

## C.2 SENSITIVITY ANALYSIS OF SAMPLE SIZE

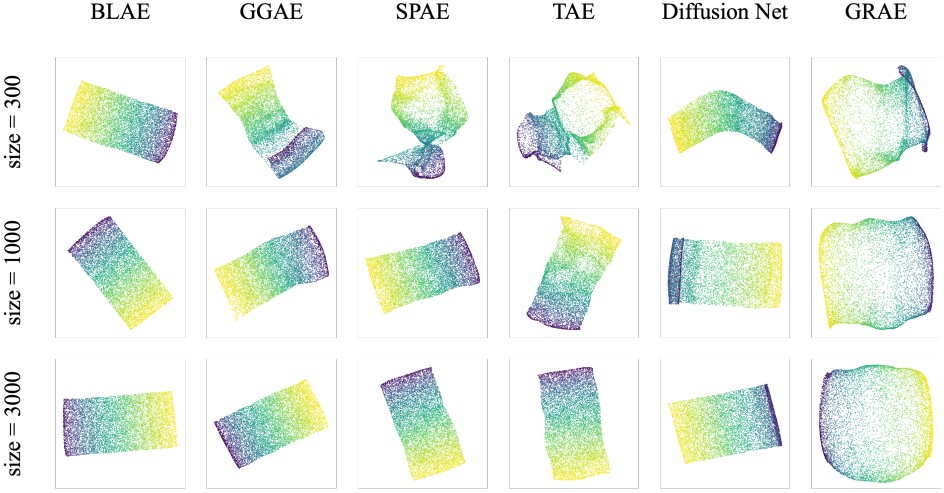

Figure 8: 2-D latent representations learned by BLAE and graph-based baselines trained with different sample sizes (400, 1000, 3000) on the Swiss Roll data. All models were trained on the indicated sample sizes, while visualizations use the full set of 10,000 data points.

The performance of graph-based methods is highly sensitive to sample density, as the quality of the neighborhood graph—and hence the accuracy of geodesic distance estimation—directly depends on the number of training points. In this section, we systematically analyze how the training sample size affects the behavior of graph-based autoencoders.

Table 9: Evaluation metrics (over 5 runs) of BLAE and graph-based baselines trained with different sample sizes (400, 1000, 3000) on the Swiss Roll data. For MSE and KL metrics, lower values are better; for $k$-NN, higher values are better. The best performance for each metric is shown in bold.

| Measure | BLAE | GGAE | SPAE | TAE | Diffusion Net | GRAE |
|---|---|---|---|---|---|---|
| | | | **Sample size = 400** | | | |
| MSE($\downarrow$) | **1.52e-03$\pm$1.07e-04** | 9.69e-02$\pm$7.98e-03 | 1.86e-02$\pm$6.07e-03 | 5.39e-02$\pm$2.96e-03 | 1.34e-01$\pm$2.84e-02 | 1.80e-01$\pm$3.93e-03 |
| $k$-NN($\uparrow$) | **9.19e-01$\pm$3.10e-03** | 4.55e-01$\pm$2.89e-02 | 7.30e-01$\pm$2.38e-02 | 6.81e-01$\pm$4.18e-03 | 5.73e-01$\pm$3.05e-02 | 4.76e-01$\pm$6.31e-03 |
| KL$_{0.01}$($\downarrow$) | **2.49e-03$\pm$2.25e-04** | 1.14e-01$\pm$1.08e-02 | 2.39e-02$\pm$4.56e-03 | 3.01e-02$\pm$1.52e-03 | 4.41e-02$\pm$1.23e-02 | 1.01e-01$\pm$3.86e-03 |
| KL$_{0.1}$($\downarrow$) | **3.55e-04$\pm$3.89e-05** | 1.81e-02$\pm$1.84e-03 | 5.85e-03$\pm$9.64e-04 | 7.37e-03$\pm$3.53e-04 | 3.83e-02$\pm$4.01e-03 | 2.29e-02$\pm$1.56e-04 |
| KL$_1$($\downarrow$) | **1.78e-05$\pm$2.22e-06** | 1.52e-03$\pm$6.69e-05 | 2.57e-04$\pm$1.94e-04 | 2.56e-04$\pm$6.34e-06 | 2.21e-03$\pm$7.78e-05 | 2.30e-03$\pm$7.54e-05 |
| Runtime (s) | 8.28e+01$\pm$5.82e-01 | 1.32e+02$\pm$9.88e-01 | 7.79e+01$\pm$2.10e+00 | 1.83e+02$\pm$1.96e+00 | 9.24e+01$\pm$1.57e+00 | 7.55e+01$\pm$1.18e+00 |
| | | | **Sample size = 1000** | | | |
| MSE($\downarrow$) | **8.55e-05$\pm$7.64e-06** | 3.96e-02$\pm$1.18e-02 | 3.01e-04$\pm$6.52e-05 | 1.72e-03$\pm$4.61e-04 | 2.42e-01$\pm$3.95e-02 | 5.64e-03$\pm$1.50e-03 |
| $k$-NN($\uparrow$) | **9.81e-01$\pm$2.06e-03** | 4.87e-01$\pm$4.48e-02 | 9.35e-01$\pm$4.40e-03 | 7.16e-01$\pm$2.59e-03 | 5.03e-01$\pm$4.41e-02 | 6.31e-01$\pm$2.97e-02 |
| KL$_{0.01}$($\downarrow$) | **1.07e-04$\pm$4.28e-05** | 1.59e-01$\pm$1.72e-02 | 1.29e-03$\pm$2.95e-04 | 2.22e-03$\pm$7.23e-04 | 3.75e-02$\pm$1.41e-02 | 6.31e-02$\pm$2.97e-02 |
| KL$_{0.1}$($\downarrow$) | **1.14e-05$\pm$3.96e-06** | 2.28e-02$\pm$3.33e-03 | 2.02e-04$\pm$8.80e-05 | 2.22e-03$\pm$7.23e-04 | 3.05e-02$\pm$4.70e-03 | 1.24e-02$\pm$8.16e-03 |
| KL$_1$($\downarrow$) | **8.63e-07$\pm$3.32e-07** | 1.43e-03$\pm$9.01e-05 | 8.82e-06$\pm$4.38e-06 | 2.22e-03$\pm$7.23e-04 | 2.04e-03$\pm$1.75e-04 | 1.00e-03$\pm$6.53e-04 |
| Runtime (s) | 1.13e+02$\pm$4.14e+00 | 2.17e+02$\pm$3.45e+00 | 1.02e+02$\pm$3.70e+00 | 3.05e+02$\pm$3.68e+00 | 1.18e+02$\pm$4.39e+00 | 9.85e+01$\pm$4.99e-01 |
| | | | **Sample size = 3000** | | | |
| MSE($\downarrow$) | **6.42e-05$\pm$6.57e-06** | 1.50e-02$\pm$1.17e-02 | 1.29e-04$\pm$2.66e-05 | 5.72e-04$\pm$1.28e-04 | 2.96e-01$\pm$3.01e-03 | 3.16e-03$\pm$3.15e-04 |
| $k$-NN($\uparrow$) | **9.81e-01$\pm$3.32e-03** | 6.50e-01$\pm$7.56e-02 | 9.53e-01$\pm$4.06e-03 | 9.31e-01$\pm$6.14e-03 | 4.37e-01$\pm$4.87e-03 | 6.79e-01$\pm$3.24e-02 |
| KL$_{0.01}$($\downarrow$) | **4.49e-04$\pm$2.25e-05** | 8.68e-02$\pm$7.70e-02 | 7.03e-04$\pm$1.94e-04 | 2.49e-03$\pm$2.25e-04 | 2.07e-02$\pm$1.86e-03 | 9.30e-02$\pm$5.89e-03 |
| KL$_{0.1}$($\downarrow$) | **5.75e-05$\pm$1.01e-05** | 2.70e-02$\pm$8.78e-03 | 8.28e-05$\pm$2.45e-05 | 3.55e-04$\pm$3.89e-05 | 2.77e-02$\pm$1.76e-03 | 8.81e-03$\pm$9.79e-04 |
| KL$_1$($\downarrow$) | **1.98e-06$\pm$1.88e-07** | 1.31e-03$\pm$1.12e-03 | 3.34e-06$\pm$3.89e-07 | 1.78e-05$\pm$2.22e-06 | 2.08e-03$\pm$8.74e-05 | 7.29e-04$\pm$6.71e-05 |
| Runtime (s) | 1.88e+02$\pm$1.47e+00 | 1.97e+03$\pm$1.72e+01 | 1.82e+02$\pm$1.42e+00 | 2.16e+03$\pm$1.00e+02 | 1.94e+02$\pm$1.75e+00 | 3.37e+02$\pm$8.01e+00 |

(a) Effect of $\kappa$ on different metrics    (b) Effect of $\epsilon$ on different metrics

Figure 9: Performance evaluation across different hyperparameter settings. (a) shows the impact of varying $\epsilon$ on $k$-NN accuracy and error metrics. (b) demonstrates the effect of $\kappa$ on the same metrics. Error metrics are displayed on a logarithmic scale.

We generated 10,000 Swiss Roll samples using fixed parameters ($b = 0.15$, latent domain $[-2, 10] \times [0, 6]$), and trained models using subsets of 400, 1000, and 3000 samples. To maintain consistency and isolate the effect of sample size, we avoided removing any subregions from the latent space, ensuring a smooth geodesic-Euclidean correspondence.

As shown in Figure 8, graph-based baselines exhibit increasing distortion in the latent representations as the sample size decreases, reflecting their reliance on high-resolution graphs for structural accuracy. In contrast, BLAE consistently preserves the underlying geometry across all sample sizes, demonstrating strong robustness to data sparsity and limited supervision.

### C.3 HYPERPARAMETER SENSITIVITY ANALYSIS

To assess robustness to hyperparameter choices, we vary the bi-Lipschitz constant $\kappa$ and separation threshold $\epsilon$ on Swiss Roll, keeping other settings fixed. Detailed discussion and ablation studies on individual regularization terms and latent space visualizations are provided in Appendix C.4 and C.5.

**Bi-Lipschitz constant $\kappa$.** Figure 9a shows performance as $\kappa \in \{1.0, 1.1, 1.2, 1.5, 2.0, 5.0, 10.0\}$. k-NN recall peaks at $\kappa = 1.2$, demonstrating optimal balance between geometric preservation and flexibility. Overly strict constraints ($\kappa = 1.0$) enforce near-isomery, but slight relaxation maintains high fidelity while improving robustness. Excessive relaxation approaches unconstrained behavior and causes significant deterioration. Notably, MSE remains stable across all $\kappa$, confirming that geometric preservation drives the performance trade-off. $\kappa \in [1.0, 1.2]$ provides robust performance.

**Separation threshold $\epsilon$.** Figure 9b shows results for $\epsilon \in \{0.2, 0.3, \ldots, 0.8\}$. k-NN recall peaks at small $\epsilon$ values and declines gracefully as $\epsilon$ increases. This behavior reflects two competing effects: smaller $\epsilon$ provides flexible separation that tolerates geodesic approximation errors, while larger $\epsilon$ enforces increasingly strict separation constraints. As $\epsilon$ approaches 1, the constraint increasingly resembles rigid distance-preserving requirements similar to SPAE, which suffers from vulnerability to geodesic approximation errors and conflicts with bi-Lipschitz relaxation. KL divergence increases approximately one order of magnitude across this range. The optimal range $\epsilon \in [0.2, 0.4]$ balances effective topological separation with robustness to distance estimation errors.

### C.3.1 LATENT VISUALIZATIONS FOR HYPERPARAMETER SENSITIVITY

**Bi-Lipschitz constant $\kappa$.** Figure 10 shows latent embeddings across different $\kappa$. For small values ($\kappa \in [1.0, 1.2]$), the Swiss Roll structure is excellently preserved with smooth rectangular boundaries and uniform point density. At moderate values ($\kappa \in [1.5, 2.0]$), the overall structure remains intact, but subtle irregularities begin to appear along the manifold boundaries. As $\kappa$ increases beyond 2.0, the geometric distortion becomes progressively more pronounced: the rectangular boundaries lose regularity and the manifold exhibits significant warping with visible curvature in regions that should be flat. Notably, even at extreme values, the topological correctness is maintained—the manifold remains properly unfolded without collapse. This visual degradation directly corresponds to the quantitative decline in $k$-NN recall shown in Figure 9a. The visualizations confirm that while the bi-Lipschitz constraint is admissible (allowing geometric relaxation), tighter bounds ($\kappa$ closer to 1) better preserve the intrinsic manifold geometry.

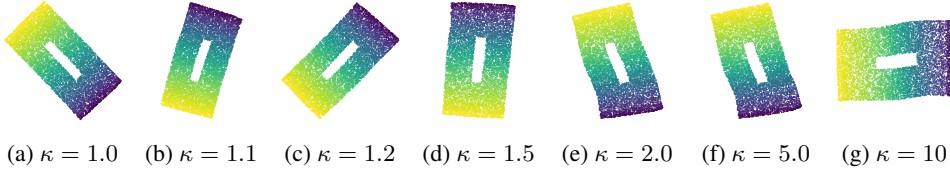

(a) $\kappa = 1.0$    (b) $\kappa = 1.1$    (c) $\kappa = 1.2$    (d) $\kappa = 1.5$    (e) $\kappa = 2.0$    (f) $\kappa = 5.0$    (g) $\kappa = 10$

Figure 10: Sensitivity analysis of $\kappa$: 2-D latent representation of Swiss Roll data learned by BLAE with different $\kappa$ values.

**Separation threshold $\epsilon$.** Figure 11 visualizes how latent structure evolves as $\epsilon$ varies from 0.2 to 0.8. Low $\epsilon$ values (0.2–0.4) produce consistently high-quality embeddings that faithfully preserve the rectangular structure of the Swiss Roll's ground truth parameterization. The manifold boundaries are smooth and regular, and the three embeddings are visually nearly indistinguishable. This stability confirms that performance is robust within this range. Moderate $\epsilon$ values (0.5–0.6) show the emergence of subtle geometric distortions. The rectangular boundaries become slightly less regular, though overall topology remains correct. High $\epsilon$ values (0.7–0.8) exhibit significant geometric irregularities. The rectangular structure is distorted, and local smoothness is compromised. However, critically, even at $\epsilon = 0.8$, the manifold topology is still preserved—no collapse occurs, and distant manifold regions remain separated. This graceful degradation aligns with the quantitative $k$-NN decline shown in Figure 9b. The visualizations confirm that smaller $\epsilon$ values provide flexible separation that tolerates geodesic approximation errors, while larger $\epsilon$ values enforce increasingly rigid constraints. As $\epsilon$ grows, the requirement $\frac{d_{\mathcal{N}}(\mathcal{E}_\theta(x), \mathcal{E}_\theta(y))}{d_{\mathcal{M}}(x,y)} > \epsilon$ for points satisfying $d_{\mathcal{M}}(x,y) \geq \delta$ becomes more restrictive, approaching strict distance preservation that is vulnerable to estimation errors, leading to the geometric drift visible at larger $\epsilon$ values.

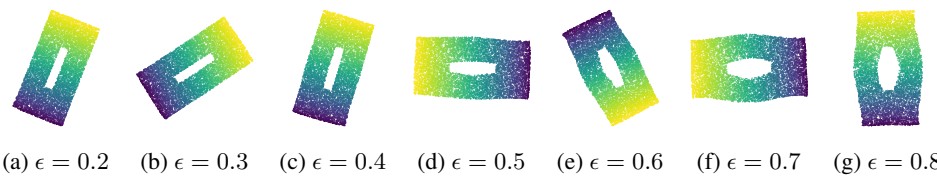

(a) $\epsilon = 0.2$    (b) $\epsilon = 0.3$    (c) $\epsilon = 0.4$    (d) $\epsilon = 0.5$    (e) $\epsilon = 0.6$    (f) $\epsilon = 0.7$    (g) $\epsilon = 0.8$

Figure 11: Sensitivity analysis of $\epsilon$: 2-D latent representation of Swiss Roll data learned by BLAE with different $\epsilon$ values.

### C.4 Comparison of injective regularization and SPAE regularizations

Our injective regularization shares structural similarities with SPAE's regularization in their formal use of geodesic distance ratios between the data manifold $\mathcal{M}$ and the latent space $\mathcal{N}$. Both frameworks impose constraints involving $d_\mathcal{M}/d_\mathcal{N}$. However, their underlying philosophies differ significantly.

SPAE enforces a strict constraint of $d_\mathcal{M}/d_\mathcal{N} = $ constant, relying on a graph-based approximation $\hat{d}_\mathcal{M}$ that introduces systematic bias under finite sampling. Additionally, SPAE substitutes $d_\mathcal{N}$ with the Euclidean norm $\|\cdot\|_2$, which inherently underestimates geodesic distances—especially in non-convex latent spaces, where equality $\|\cdot\|_2 = d_\mathcal{N}$ holds only in convex settings. For example, in the Swiss Roll dataset, points located on opposite sides of the removed strip exhibit large geodesic distances, yet are deceptively close in Euclidean space.

In contrast, our injective regularization selectively penalizes extreme distortions in the $d_\mathcal{M}/d_\mathcal{N}$ ratio, and is inherently more tolerant to approximation errors. Rather than enforcing a global constraint, it activates only when a specific geometric violation is detected, enabling targeted regularization through appropriate gradient signals.

Notably, combining SPAE with gradient-based (bi-Lipschitz) regularization results in inferior performance. This degradation arises from the compounded effects of distance approximation errors and the rigid constraints enforced by SPAE.

### C.5 Ablation Studies

#### C.5.1 Combining Injective Regularization with Gradient-Based Methods

To demonstrate that injective regularization can alleviate pathological local minima in existing gradient-based autoencoders, we combine our separation criterion with two representative gradient-based regularizers: CAE and GAE. We conduct experiments on the Swiss Roll dataset with 1,500 training samples and visualize using the complete 10,000-point dataset.

Figure 12 shows the results of combining injective regularization with CAE and GAE. Panels (a)-(b) show that CAE and GAE alone struggle to properly unfold the Swiss Roll manifold, exhibiting non-injective collapse similar to the vanilla autoencoder. However, when combined with our injective regularization (panels (c)-(d)), both methods successfully preserve the manifold topology and properly unfold the structure without collapse. While the unfolded representations show some geometric irregularities along the boundaries compared to the ideal rectangular structure, the critical topological property, preventing distant manifold regions from collapsing into nearby latent codes, is successfully enforced. This demonstrates that our injective regularization acts as a complementary constraint that helps gradient-based methods escape pathological local minima caused by non-injectivity. For optimal geometric preservation, including smooth boundaries, combining both injective and bi-Lipschitz regularization (as in full BLAE) is necessary, which we demonstrate throughout the next section.

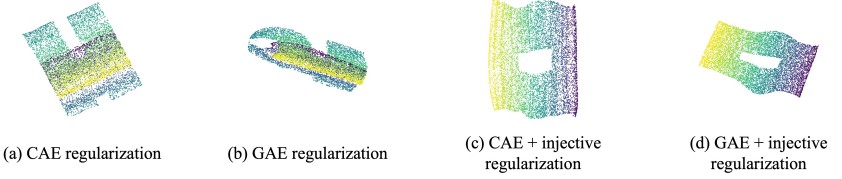

(a) CAE regularization     (b) GAE regularization     (c) CAE + injective regularization     (d) GAE + injective regularization

Figure 12: 2-D latent representation of Swiss Roll data learned by autoencoders with (a) only CAE regularization, (b) only GAE regularization, (c) CAE + injective regularizations, (d) CAE + injective regularizations.

#### C.5.2 Complementary Roles of Injective and Bi-Lipschitz Regularization

To validate that both regularization terms serve distinct and complementary purposes, we conduct ablation experiments on the Swiss Roll dataset. Figure 13 presents 2-D latent representations learned under four different regularization schemes, all trained with 1500 samples and visualized using the complete 10,000-point dataset.

**Injective regularization only.** (Fig. 13 (a)) With $\lambda_{\text{reg}} > 0$ and $\lambda_{\text{bi-Lip}} = 0$, the model successfully unfolds the Swiss Roll topology, confirming that the separation criterion eliminates non-injective collapse. However, the embedding exhibits geometric irregularities and uneven point density. This demonstrates that injectivity alone is necessary but insufficient for smooth geometric preservation.

**SPAE regularization only.** (Fig. 13 (b)) SPAE's distance-preserving constraint produces a cleaner rectangular embedding with more uniform point distribution. However, as discussed in Appendix C.4, SPAE's rigid constraint $d_{\mathcal{M}}/d_{\mathcal{N}} = \text{constant}$ is vulnerable to geodesic distance approximation errors, especially near the removed strip region.

**Injective + bi-Lipschitz regularization (BLAE).** (Fig. 13 (c)) Our full framework combines both terms with $\kappa = 1$, achieving: (1) topological correctness from the injective term, and (2) geometric smoothness from the bi-Lipschitz term. The embedding closely resembles the ground truth parameterization with a uniform point distribution and regular boundaries.

**SPAE with bi-Lipschitz regularization.** (Fig. 13 (d)) This combination yields inferior results with visible distortion and irregular geometry. This validates our analysis in Appendix C.4: SPAE's strict global constraint conflicts with bi-Lipschitz relaxation, and approximation errors compound. In contrast, our injective regularization only penalizes extreme violations, making it naturally compatible with bi-Lipschitz constraints.

**Bi-Lipschitz regularization only.** (Fig. 13 (e)) The model fails to unfold the Swiss Roll.

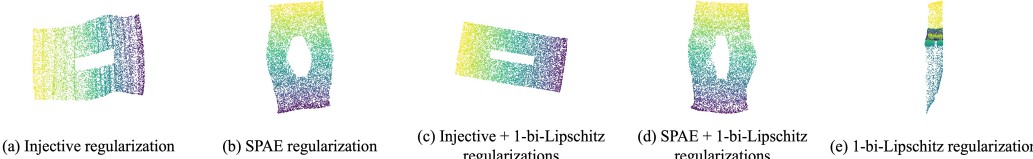

(a) Injective regularization  (b) SPAE regularization  (c) Injective + 1-bi-Lipschitz regularizations  (d) SPAE + 1-bi-Lipschitz regularizations  (e) 1-bi-Lipschitz regularization

Figure 13: 2-D latent representation of Swiss Roll data learned by autoencoders with (a) only injective regularization, (b) only SPAE regularization, (c) injective + 1-bi-Lipschitz regularizations, (d) SPAE + 1-bi-Lipschitz regularizations, (e) only 1-bi-Lipschitz regularization.

## C.6 EXTRA DATASETS AND EXPERIMENTS

### C.6.1 SSREAD DATABASE

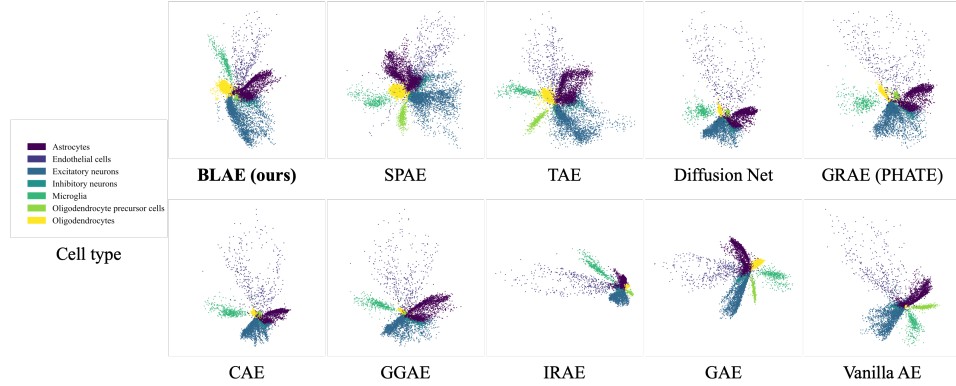

Figure 14: Left: Single cell type in the AD00109 dataset from the ssREAD database. *Others:* 2-D latent representations learned by BLAE and other baseline methods.

We utilize the AD00109 dataset from the ssREAD database, which provides single-nucleus RNA sequencing (snRNA-seq) data from brain tissue of Alzheimer's disease (AD) patients. This dataset contains transcriptomic profiles from the prefrontal cortex of several female AD patients, aged between 77 and over 90. Sequencing was performed using the 10x Genomics Chromium platform.

Standard preprocessing steps were applied, including quality control, normalization, dimensionality reduction, and unsupervised clustering. The resulting dataset consists of 9,891 cells and 27,801 genes, annotated into seven distinct cell types.

For downstream analysis, we apply principal component analysis (PCA) and retain the top 50 principal components. We split the data by using 25% of the cells for training and the remaining 75% for evaluation. Figure 14 shows the 2D latent representations learned by BLAE and several baseline methods. Among these, BLAE, SPAE, TAE, and GAE better preserve the distinct boundaries between different cell types. To quantitatively assess representation quality, we train a Support Vector Machine (SVM) classifier on the latent representations (latent dimension $= 2, 32$) of the training set and report classification accuracy on the evaluation set. Results are summarized in Table 10, where BLAE achieves the highest accuracy on both latent dimensions, indicating that it learns the most discriminative latent representations for single-cell type identification.

Table 10: Classification accuracy (over 5 runs) of SVM trained on latent representations learned by different models. BLAE achieves the highest accuracy, demonstrating better preservation of cell-type-specific information in the latent space. The best performance is shown in bold.

| Model | Latent dimension = 2 | Latent dimension = 32 |
|---|---|---|
| BLAE | $\mathbf{0.9250 \pm 0.0033}$ | $\mathbf{0.9626 \pm 0.0013}$ |
| SPAE | $0.9146 \pm 0.0206$ | $0.9576 \pm 0.0019$ |
| TAE | $0.9107 \pm 0.0121$ | $0.9524 \pm 0.0026$ |
| DN | $0.9167 \pm 0.0064$ | $0.4200 \pm 0.0821$ |
| GRAE | $0.9131 \pm 0.0119$ | $0.9476 \pm 0.0022$ |
| CAE | $0.9222 \pm 0.0030$ | $0.9517 \pm 0.0025$ |
| GGAE | $0.8855 \pm 0.0161$ | $0.9556 \pm 0.0017$ |
| IRAE | $0.9066 \pm 0.0200$ | $0.9605 \pm 0.0016$ |
| GAE | $0.9185 \pm 0.0113$ | $0.9575 \pm 0.0022$ |
| Vanilla AE | $0.8996 \pm 0.0172$ | $0.9557 \pm 0.0017$ |

### C.6.2 EXTENSION TO VARIATIONAL AUTOENCODERS

To demonstrate that our regularization framework generalizes beyond deterministic autoencoders, we extend BLAE to variational autoencoders (VAEs), creating Bi-Lipschitz VAE (BL-VAE). This validates that our theoretical insights apply to probabilistic latent variable models. We evaluate on the Swiss Roll dataset from Section 5 with identical settings, applying our regularization (equation BLAE) to the VAE decoder. We assess performance using four metrics: Reconstruction MSE (fidelity), Fréchet Inception Distance (FID) (generation quality via feature distribution comparison), KL Divergence (posterior-prior divergence in VAE objective), and Mutual Information Gap (MIG) (disentanglement quality).

Table 11 shows BL-VAE achieves substantially lower reconstruction error (7.09e-2 vs. 1.42e-1), confirming our regularization helps escape pathological local minima in probabilistic settings. FID scores are comparable. While BL-VAE exhibits higher KL divergence due to geometric constraints, it achieves dramatically better disentanglement (MIG: 0.828 vs. 0.534), indicating the latent space better aligns with true underlying factors of variation. These results confirm our framework successfully extends to variational settings while preserving its core benefits.

Table 11: Swiss Roll Results (Mean $\pm$ Std over 5 seeds)

| Model | Recon MSE ↓ | FID ↓ | KL | MIG ↑ |
|---|---|---|---|---|
| VAE | $1.42\text{e-}1 \pm 3.65\text{e-}3$ | $1.23\text{e-}2 \pm 2.32\text{e-}3$ | $3.06\text{e+}0 \pm 1.33\text{e-}2$ | $5.34\text{e-}1 \pm 1.96\text{e-}1$ |
| BL-VAE | $\mathbf{7.09\text{e-}2 \pm 1.41\text{e-}3}$ | $\mathbf{1.15\text{e-}2 \pm 1.15\text{e-}3}$ | $5.58\text{e+}0 \pm 4.06\text{e-}2$ | $\mathbf{8.28\text{e-}1 \pm 1.93\text{e-}2}$ |

### C.6.3 TOY EXAMPLE WITH INCREASED SAMPLE DENSITY

To address concerns about whether non-injective embeddings reflect genuine inadequacy or merely insufficient data, we extend the toy example from Figure 1 by increasing the sample size from 20 to 200 points while maintaining the same V-shaped manifold structure.

We sample 200 points uniformly from the V-shaped manifold and train both a vanilla autoencoder and BLAE with identical architectures (hidden dimension = 16). Both models are trained under the same conditions to isolate the effect of injective regularization. As shown in Figure 15, even with 10× more training samples, the vanilla autoencoder (panels c-d) continues to exhibit non-injective collapse: the latent representation shows severe overlap between the red and blue classes, resulting in distorted reconstruction with an irregular curve. In contrast, BLAE (panels e-f) achieves proper class separation in the latent space and accurately reconstructs the V-shaped structure.

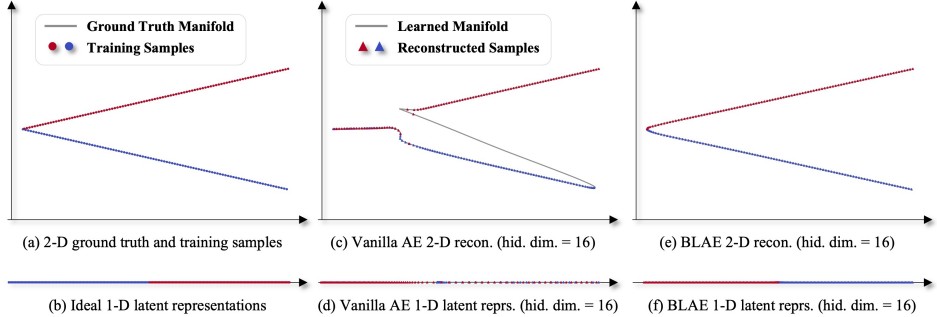

Figure 15: Extended toy example demonstrating the non-injective encoder bottleneck. (a) 200 training points sampled from a V-shaped manifold with two classes (red and blue). (b) Ground truth: an ideal encoder separates the two classes in a 1D latent space. (c) Reconstructed manifolds by Vanilla AE with hidden dimension 16. (d) Corresponding latent representations learned by Vanilla AE. (e) Reconstructed manifolds by BLAE with hidden dimension 16. (f) Corresponding latent representations learned by BLAE show proper class separation.

## D  USAGE OF LARGE LANGUAGE MODELS (LLMS)

We used large language models as a general-purpose writing assistant. Its role was limited to grammar checking, minor stylistic polishing, and improving the clarity of phrasing in some parts of the manuscript. The authors made all substantive contributions to the research and writing.

