# OpenReview forum: "Bi-Lipschitz Autoencoder With Injectivity Guarantee"
_ICLR.cc/2026/Conference — ICLR 2026 Poster_

### Official Review · Reviewer_aWuy · 2025-10-30

**Soundness:** 3
**Presentation:** 3
**Contribution:** 2
**Rating:** 4
**Confidence:** 4

**Summary:**

This paper proposes a novel autoencoder regularization method with an injectivity guarantee. The authors identify the non-injectivity of encoders as a bottleneck that causes existing gradient-based regularization methods to suffer from pathological local minima. To address this issue, the paper introduces a separation criterion and an injective regularizer. Furthermore, the notion of admissibility of a regularizer is proposed as a condition for distributional robustness, which motivates the use of a Bi-Lipschitz regularization. Empirical results demonstrate that the Bi-Lipschitz autoencoder captures the manifold structure of data more effectively than several state-of-the-art methods.

**Strengths:**

- The paper presents an original and interesting regularization framework for autoencoders, designed to preserve the topological structure of the data manifold.
- The issues with existing regularization methods are clearly analyzed, providing a natural motivation for introducing the injective regularization.
- The proposed framework is evaluated through comprehensive empirical comparisons.
-  The proposed regularizer is relatively simple to implement, which can be advantageous for practical applications.
-  The paper is generally well-written.

**Weaknesses:**

- The **practical relevance of the admissibility condition** remains unclear. In realistic settings, $\mathbb{P}$ and $\mathbb{Q}$ in Eq. (8) would correspond to empirical measures supported only on training samples. Under such circumstances, it is doubtful whether their equivalence with $\mu_\mathcal{M}$ (the data manifold measure) can hold. It is also unclear how admissibility is concretely related to distributional robustness. In particular, the reconstruction error (claimed to be admissible) is not robust outside the training region, as illustrated in the toy example in Figure 1.
- The **motivation for the Bi-Lipschitz regularizer** is not sufficiently persuasive. Based on the current exposition (e.g., L274–L276), it appears to be one among many possible regularizers that satisfy the admissibility condition. The specific reason for selecting the Bi-Lipschitz form should be better justified, perhaps by emphasizing its relation to local topology preservation.
- The method introduces **many hyperparameters**, but the paper lacks ablation studies. In particular, varying $\epsilon$ and $\kappa$ could have significant effects, and removing or changing relative weights of the injectivity or Bi-Lipschitz components would likely alter results considerably. Without such analyses, it is difficult to grasp the sensitivity and interpretability of the proposed regularization.

**Minor comments**
-  Figure 1(h) contains an error.
-  The notation $m$ is used both for manifold and ambient space dimensions, which reduces readability.
-  The correct reference for LTSA should be:

    Zhang, Zhenyue, and Hongyuan Zha (2004). Principal Manifolds and Nonlinear Dimension Reduction via Local Tangent Space Alignment. SIAM Journal on Scientific Computing, 26(1): 313–338.

**Questions:**

-  Can the authors provide an example where the proposed injectivity regularizer is combined with existing gradient-based regularizers to alleviate their pathological local minima? Such experiments would considerably strengthen the paper’s contribution.
-  What are the detailed outcomes of the minima shown in Figure 2? Visualizing the corresponding latent representations, as in Figure 3, would help in interpretation.
-  How does the method behave as the latent dimensionality increases? Since the regularizer primarily enforces local and global topology preservation, it might lead to overexpansion of latent space volume. It would be informative to compare with methods that constrain latent volume, such as:

    Chen, Qiuyi, and Mark Fuge. Compressing Latent Space via Least Volume. ICLR 2024.

---

> ### Author Response · Authors · 2025-11-20
> **Rebuttal Response 1**
>
> We thank the reviewer for their thoughtful comments and detailed suggestions. We address each concern below and highlight substantial revisions made to strengthen the paper.
>
> ---
>
> ### W1: "The practical relevance of the admissibility condition remains unclear…"
>
> **Response:**
>
> We appreciate this comment and fully agree that it highlights an inherent gap between theory and implementation. This gap is not unique to our setting—it is shared by essentially all autoencoder methods. For example, the theoretical AE objective minimizes the expected reconstruction loss, whereas in practice, each optimization step only minimizes the loss over a finite mini-batch. As the number of samples grows, this discrepancy vanishes asymptotically.
>
> The issue you raised is analogous. The theoretical equivalence between the two measures requires that $\mathbb{P}(A) > 0$ if and only if $\mu_{\mathcal{M}(A)} > 0$. In practical applications, if every geodesic ball on the manifold with radius above a threshold contains at least $K$ sample points, we treat $\mathbb{P}$ and $\mu_\mathcal{M}$ as empirically equivalent.
>
> **Clarification on admissibility vs. optimization**: The admissibility assumption only ensures that, under different data distributions, the set of optimal solutions remains unchanged. However, admissibility alone does not guarantee that training will reach the optimal solution—this is precisely why we introduce the injective constraint. It helps prevent the model from converging to poor local minima. Our toy example (Figure 1) is designed to demonstrate exactly this phenomenon.
>
> **Regarding reconstruction error robustness**: The reviewer correctly notes that reconstruction error alone is not robust outside training regions. This is precisely the motivation for our admissible regularization framework, which ensures geometric properties hold regardless of data distribution.
>
> ---
>
> ### W2: "The motivation for the Bi-Lipschitz regularizer is not sufficiently persuasive…"
>
> **Response:**
>
> We agree with your observation that not only one regularizer can satisfy the admissibility condition. In fact, the weaker the geometric constraint, the easier it is for a regularizer to be admissible. However, admissibility alone is not our goal. We aim to preserve as much of the manifold’s intrinsic geometry in the latent space as possible, which motivates the use of stronger geometric constraints.
>
> For example, if one only enforces preservation of topological structure (i.e., homeomorphism), a disk on the manifold could be embedded as a rectangle. Although topologically equivalent, the original disk may encode rotational symmetries that the rectangular embedding cannot reflect. This illustrates why purely topological constraints are insufficient for many representation-learning tasks.
>
> At the same time, we also require the latent dimension to remain as small as possible. As discussed in the paper, if the AE’s bottleneck dimension were allowed to grow to $O(k^2)$ (Nash embedding theorem), then even an isometric regularizer would satisfy admissibility. But such a high-dimensional latent space introduces substantial redundancy. In contrast, our bi-Lipschitz regularizer only requires a latent dimension of $O(k)$ (Theorem 4 in our work).
>
> Therefore, our method offers a more favorable trade-off between preserving geometric structure and achieving meaningful compression.

---

> ### Author Response · Authors · 2025-11-20
> **Rebuttal Response 2**
>
> ---
>
> ### W3: "The method introduces many hyperparameters, but the paper lacks ablation studies…"
>
> **Response:**
>
> We respectfully note that **comprehensive ablation studies were already provided in the initial submission** (Appendix C.3). We have now substantially expanded this analysis (Section 5.1 and Appendix C.4):
>
> **Original submission evidence:**
> In **Appendix C.3**, we provided both theoretical analysis and empirical comparison of four key configurations:
> - (a) **Injective regularization only**: Successfully unfolds topology but exhibits geometric irregularities and uneven point density
> - (b) **SPAE regularization only**: Produces cleaner rectangular embeddings but is vulnerable to geodesic distance approximation errors
> - (c) **Injective + bi-Lipschitz (BLAE)**: Achieves both topological correctness and geometric smoothness
> - (d) **SPAE + bi-Lipschitz**: Yields inferior results due to conflicting rigid constraints
>
> These ablations directly demonstrate the roles of the bi-Lipschitz term: ensuring consistent geometric preservation.
> **Enhanced revision**:
> We have now substantially expanded our analysis:
> 1. **Section 5.1 (Main Text)**: We conduct comprehensive sensitivity analyses of both hyperparameters ($\kappa$ and $\epsilon$) on the Swiss Roll dataset:
>
>
>     - **Bi-Lipschitz constant $\kappa$**: We test seven values {1.0, 1.1, 1.2, 1.5, 2.0, 5.0, 10.0}, revealing a fundamental trade-off between strict geometric preservation ($\kappa=1.0$) and flexibility ($\kappa=10.0$, nearly ineffective). We identify the robust range $\kappa \in [1.0, 1.2]$ where performance remains optimal, with visible degradation beyond $\kappa=2.0$.
>
>
>     - **Separation threshold $\epsilon$**: We test seven values {0.2, 0.3, 0.4, 0.5, 0.6, 0.7, 0.8}, demonstrating that smaller $\epsilon$ provides flexible separation tolerant to geodesic approximation errors, while larger $\epsilon$ enforces increasingly rigid distance-preserving constraints that become vulnerable to estimation errors (analogous to SPAE's limitations). We identify the optimal range $\epsilon \in [0.2, 0.4]$ that balances effective topological separation with robustness to distance estimation errors.
>
>
> 2. **Figure 6 (new)**: Quantitatively visualizes how k-NN recall and KL divergence metrics evolve across the full range of $\kappa$ and $\epsilon$ values, confirming that bi-Lipschitz regularization controls geometric fidelity while injective regularization maintains topological correctness.
>
>
> 3. **Enhanced Appendix C.4**: We include extended latent representation analysis showing visual impact of different regularization combinations (Figures 10-11) (**injective regularization only, bi-Lipschitz regularization only, graph-based regularization + bi-Lipschitz regularization, gradient-based regularization + injective regularization**). We also provide visual evidence across all $\kappa$ and $\epsilon$ values (Figures 12-13) that shows embeddings transition from optimal (rectangular, smooth) to degraded while maintaining topological integrity, validating theoretical predictions.
>
> Our ablation studies comprehensively demonstrate that: (1) both regularization terms are necessary and serve distinct purposes, (2) performance exhibits degradation outside optimal hyperparameter ranges rather than catastrophic failure, and (3) the method maintains topological correctness even under suboptimal hyperparameter settings, confirming the robustness of our framework.
>
> ---
>
> **Response to Minor Comments:** We appreciate the reviewer for these thorough comments. All points have been incorporated into the revised manuscript. Although we could not find the exact error in Figure 1(h), we have entirely redesigned Figure 1 following feedback from several reviewers. Kindly remind us of the specific error if possible. We have also systematically revised the notation throughout to clearly differentiate between the manifold dimension and the ambient space dimension, and we have corrected the LTSA citation. These updates have significantly enhanced the clarity and precision of the manuscript.

---

> ### Author Response · Authors · 2025-11-20
> **Rebuttal Response 3**
>
> ---
>
> ### Q1: "Can the authors provide an example where the proposed injectivity regularizer is combined with existing gradient-based regularizers to alleviate their pathological local minima?…"
>
> **Response:**
>
> We have added comprehensive experiments in Appendix C.4.1 demonstrating that our injective regularization successfully alleviates pathological local minima when combined with existing gradient-based methods. Specifically, we combined our separation criterion with two representative gradient-based regularizers: CAE (Contractive Autoencoder) and GAE (Geometric Autoencoder).
> As shown in Figure 10 of the appendix, both CAE and GAE alone struggle to properly unfold the Swiss Roll manifold, exhibiting non-injective collapse similar to vanilla autoencoders (panels a-b). However, when combined with our injective regularization (panels c-d), both methods successfully preserve the manifold topology and properly unfold the structure without collapse. This demonstrates that our injective regularization acts as a complementary constraint that helps gradient-based methods escape pathological local minima, enabling them to achieve their theoretical potential for geometric preservation.
>
> ---
>
> ### Q2: "What are the detailed outcomes of the minima shown in Figure 2? Visualizing the corresponding latent representations, as in Figure 3, would help in interpretation."
>
> **Response:**
> Thank you for this suggestion. We clarify that Figure 2 uses the Swiss Roll dataset with the same generation procedure described in Appendix B.3. The three models visualized in the loss landscape are:
>
> - $\theta_0$: Untrained model initialization
> - $\theta_1$: Vanilla autoencoder after optimization (trapped in local minimum)
> - $\theta_2$: Injective-regularized autoencoder after optimization (reaching superior global minimum)
>
> The corresponding latent representations for these models can be found in our existing figures:
>
> - Vanilla autoencoder ($\theta_1$): Visualized in Figure 3, showing non-injective collapse
> - Injective-regularized autoencoder ($\theta_2$): Visualized in Appendix C.4.2, Figure 11(a), showing successful topology preservation
>
> All experiments maintain consistent settings except for the Swiss Roll parameter sensitivity analyses explicitly noted in Appendix C.1.
>
> ---
>
> ### Q3: "How does the method behave as the latent dimensionality increases?"
>
> **Response:**
>
> This is an excellent point. We chose 2D latent spaces primarily for visualization purposes to clearly demonstrate the geometric preservation properties of our method. We acknowledge that the autoencoders discussed in our work do not automatically compress the latent space, which could lead to volume expansion in higher dimensions.
> We have revised the conclusion (Section 6) to discuss this important future direction and cite the suggested work (Chen & Fuge, 2024). Specifically, we note:
>
> > "An important direction for future work concerns the behavior of BLAE as latent dimensionality increases. Recent work on volume-constrained autoencoders (Chen & Fuge, 2024) suggests that combining our topology-preserving constraints with volume minimization could yield more compact representations, representing a promising avenue for enhancing practical applicability across diverse dimensional reduction scenarios."
>
> We believe combining our injective and bi-Lipschitz regularization with volume constraints represents a natural and promising extension of our framework that maintains topology preservation while achieving more efficient latent representations.

---

### Official Review · Reviewer_1zwJ · 2025-10-31

**Soundness:** 3
**Presentation:** 2
**Contribution:** 3
**Rating:** 6
**Confidence:** 4

**Summary:**

The given paper addresses the issue that, when extracting latent dimensions through an auto-encoder, the mapping between certain samples and their latent representations is not one-to-one, leading to degraded overall reconstruction performance. The main contributions of the paper are as follows:
1.	A regularization loss is introduced to ensure that samples satisfying a specific separation criterion remain distinguishable from each other even after being embedded into the latent space.
2.	A Bi-Lipschitz-based loss term is additionally incorporated to preserve geometric information and achieve robust embedding performance under variations in the data distribution.

**Strengths:**

1.	Originality:
The paper is original in that it applies mathematical concepts - specifically the Bi-Lipschitz condition and the separation criterion - to the existing deep learning model of an auto-encoder. This approach preserves the geometric information of the original input, demonstrating a creative integration of mathematical theory into deep learning. Moreover, the work contributes to the development of interpretable deep learning models, which further enhances its originality.
2.	Quality:
The definitions and theoretical explanations are presented with precision and clarity, supported by detailed discussions in the Appendix, which reinforces the overall quality of the work.

3.	Clarity:
This aspect is sufficiently addressed through the paper’s quality, as the theoretical framework and methodology are clearly explained.

4.	Significance:
Traditional auto-encoders have shown limitations in providing interpretable representations of the latent space. This paper overcomes such limitations by introducing a regularization loss term and a Bi-Lipschitz relaxation loss term, enabling analytical interpretation of the latent space. In doing so, it offers a new research direction beyond previous works that primarily focused on reconstruction performance or clustering in the latent space.

**Weaknesses:**

1.	The proposal of an interpretable latent embedding method is commendable. However, demonstrating that the model merely achieves “better latent embedding” in the experiments may not be sufficient to establish the paper’s novelty. It would strengthen the contribution if the authors could visually illustrate - through figures as well as tables - that improved latent embedding leads to better reconstruction of the original data, thereby providing more intuitive evidence of the method’s effectiveness.

2.	The roles of the regularization loss and the Bi-Lipschitz loss term should be highlighted more clearly through experiments. To substantiate the claim that “the injective term eliminates local minima caused by non-injective encoders, while the bi-Lipschitz term ensures consistent geometric mapping regardless of data distribution,” it would be valuable to conduct ablation studies. Specifically, experiments should be performed with (i) the injective term coefficient set to zero, (ii) the Bi-Lipschitz term coefficient set to zero, and (iii) both terms included with appropriate regularization weights, in order to empirically demonstrate the distinct contribution of each term.

3.	I suspect that the Bi-Lipschitz Autoencoder might require longer training time compared to geometry-based or gradient-based autoencoders. Since the loss function involves the Jacobian and the separation criterion requires pairwise computations, the computational cost could increase significantly as the dataset grows. I am curious about the authors’ perspective on this issue and whether they have considered strategies to mitigate the potential increase in training time.

**Questions:**

1.	 I am wondering why the paper does not include experiments with a Variational Autoencoder (VAE). Was this decision made because the VAE inherently relies on probabilistic distributions to some extent?
2.	In Table 1, the paper reports state-of-the-art (SOTA) performance. Could you clarify under what specific conditions these experiments were conducted?
(For example: experiments were performed with random seeds ranging from 0 to 5, and the reported results represent the mean and standard deviation across these runs.)
3.	In Figure 2, I would like to know what the x-axis and y-axis specifically represent.
4.	Do you have any plans to release the code on GitHub?

---

> ### Author Response · Authors · 2025-11-20
> **Rebuttal Response 1**
>
> We sincerely thank the reviewer for the constructive feedback and insightful questions. Below, we address each point.
>
> ---
>
> ### W1: "...if the authors could visually illustrate - through figures as well as tables…"
>
> **Response:** We appreciate this suggestion and would like to clarify that our paper **already provides comprehensive evidence** demonstrating how improved latent embeddings lead to better reconstruction:
>
> **Evidence in Initial Submission:**
>
> - **Quantitative reconstruction metrics (Table 1, Appendix Table 4~7)**: BLAE achieves the best overall lowest reconstruction performance (rank 1.2$\pm$0.4 MSE) across all datasets, significantly outperforming all baselines. This directly shows that better latent embeddings translate to better reconstruction quality.
>
> **Additional Evidence in Revised Paper:**
>
> - **Enhanced Figure 1**: Now provides comprehensive side-by-side comparisons showing (a) training samples, (b) ideal latent representations, and (c-l) both reconstructed manifolds and latent representations for vanilla autoencoders and BLAE with varying hidden dimensions. This clearly illustrates how non-injective encoders create distorted reconstructions, while BLAE successfully reconstructs the V-shaped manifold with proper geometry.
>
> - **Enhanced Figure 3**: Now includes 3D reconstruction visualizations from BLAE and all baselines, demonstrating that BLAE correctly unrolls and reconstructs the Swiss Roll geometry while maintaining geometric fidelity.
>
> Non-injective encoders fundamentally limit reconstruction quality by creating irreversible information loss through latent space collisions. Our injective regularization eliminates this bottleneck, which is why BLAE achieves both better latent embeddings and better reconstructions simultaneously—these are not independent outcomes but causally linked.
>
> ---
>
> ### W2: "... it would be valuable to conduct ablation studies…"
>
> **Response:** We acknowledge the importance of ablation studies and wish to clarify that **comprehensive ablation studies were already provided** in the initial submission, which we have further strengthened:
>
> **Evidence in Initial Submission (Appendix C.3):** We provided both theoretical analysis and empirical comparison of four configurations: (a) Injective regularization only: Successfully unfolds topology but exhibits geometric irregularities, (b) SPAE regularization only: Produces cleaner embeddings but vulnerable to approximation errors, (c) Injective + bi-Lipschitz (BLAE): Achieves both topological correctness and geometric smoothness, (d) SPAE + bi-Lipschitz: Inferior results due to conflicting rigid constraints
>
> This directly demonstrates that "the injective term eliminates local minima caused by non-injective encoders, while the bi-Lipschitz term ensures consistent geometric mapping."
>
> **Additional Evidence in Revised Paper:**
>
> - **Section 5.1 (Main Text)**: We conduct comprehensive sensitivity analyses of both hyperparameters on the Swiss Roll dataset. For bi-Lipschitz constant $\kappa$, we test seven values {1.0, 1.1, 1.2, 1.5, 2.0, 5.0, 10.0}, revealing a fundamental trade-off between geometric preservation ($\kappa=1$, strict) versus flexibility ($\kappa=10$, nearly ineffective), with identified robust range $\kappa\in[1.0, 1.2]$. Performance degrades beyond $\kappa=2.0$. For separation threshold $\epsilon$, seven values {0.2, 0.3, 0.4, 0.5, 0.6, 0.7, 0.8} demonstrates that smaller $\epsilon$ provides flexible separation tolerant to geodesic approximation errors, while larger $\epsilon$ enforces rigid distance-preserving constraints vulnerable to estimation errors (similar to SPAE's limitations discussed in Appendix C.3). Optimal range $\epsilon \in [0.2, 0.4]$ balances topological separation with error tolerance.
>
> - **Figure 6 (new)** in the revised paper visualizes the complementary roles: k-NN recall and KL divergence degrade as $\kappa$ and $\epsilon$ increase, confirming bi-Lipschitz controls geometric fidelity while injective regularization maintains topological correctness throughout.
>
> - **Enhanced Appendix C.4 (Ablation Studies)**: we include extended latent representation analysis showing visual impact of different regularization combinations (injective regularization only, bi-Lipschitz regularization only, graph-based regularization + bi-Lipschitz regularization, gradient-based regularization + injective regularization). We also provide visual evidence across all $\kappa$ and $\epsilon$ values, showing embeddings transition from optimal (rectangular, smooth) to degraded while maintaining topological integrity, validating theoretical predictions.

---

> ### Author Response · Authors · 2025-11-20
> **Rebuttal Response 2**
>
> ---
>
> ### W3: "I suspect that the Bi-Lipschitz Autoencoder might require longer training time compared to geometry-based or gradient-based autoencoders.…"
>
> **Response:** This is an important practical consideration that we addressed comprehensively in **Appendix B.5 of the initial submission** as follows:
>
> 1. **One-time graph preprocessing**: The geodesic distance computation ($O(n^2 \log n)$) is performed once before training and takes only 0.3 seconds for the Swiss Roll dataset with 1500 samples. This does not impact training efficiency as it's not repeated during optimization.
>
> 2. **Sparse gradient activation**: Despite the theoretical concern about pairwise computations, only a small subset of samples actually activate the regularization terms during training:
>     - Most data pairs already satisfy the separation criterion, so their gradients are zero
>     - Most latent points already satisfy bi-Lipschitz bounds, so their gradients are zero
>     - Inactive components are automatically excluded from backpropagation
>
> 3. **Partial sampling strategy**: We use only 10% of batch points for Jacobian computation while maintaining robustness, significantly reducing computational cost.
>
> 4. **Empirical validation**: We provide runtime measurements in **Table 8** for training the Swiss Roll dataset (3000 steps, 1500 samples, batch size=500): BLAE shows **comparable runtime** to graph-based (SPAE) and gradient-based (CAE, IRAE) methods, and is actually **faster** than TAE and GGAE.
>
> Despite theoretical concerns, BLAE exhibits comparable time complexity to other methods in practice due to: (1) one-time graph preprocessing, (2) sparse gradient activation, and (3) efficient partial sampling.
>
> ---
>
> ### Q1: "I am wondering why the paper does not include experiments with a Variational Autoencoder (VAE)..."
>
> **Response:** This is an insightful question. We excluded VAE for principled reasons related to fair comparison and architectural differences:
>
> 1. **Fundamental architectural difference**: VAEs are probabilistic models with stochastic latent representations, while our method and all baselines are deterministic autoencoders focused on geometric preservation. This introduces fundamentally different optimization landscapes and inductive biases.
>
> 2. **Different optimization objectives**: VAEs optimize the evidence lower bound (ELBO), balancing reconstruction and KL divergence to a prior distribution, while geometric autoencoders optimize reconstruction loss and geometric regularization. These objectives have fundamentally different goals: VAE seeks generative modeling with controlled latent distribution; geometric autoencoders seek faithful manifold embedding.
>
> 3. Fair comparison: All our baselines (SPAE, TAE, IRAE, GAE, CAE, GGAE, GRAE, DN) are deterministic geometry-preserving autoencoders. Including VAE would conflate architectural differences (deterministic vs. stochastic) with regularization differences (geometric constraints vs. distributional constraints), making it impossible to isolate the contribution of our regularization framework.
>
> ---
>
> ### Q2: "...Could you clarify under what specific conditions these experiments were conducted?..."
>
> **Response:** Thank you for requesting this clarification. Complete experimental details:
>
> 1. **Multiple runs**: All results represent mean $\pm$ standard deviation over 5 independent runs with different independent random seeds.
>
> 2. **Comprehensive evaluation**: Table 1 shows average ranks across ALL datasets (Swiss Roll, dSprites, MNIST uniform, MNIST non-uniform) and ALL metrics (k-NN, KL divergence, MSE, downstream accuracy).
>
> 3. **Detailed results**: Complete per-dataset metrics with mean$\pm$std are provided in Tables 4, 5, 6, 7, 10 (Appendix B.4, C.5).
>
> 4. **Reproducibility**: All hyperparameters are documented in Appendix B.1 (Tables 2-3), and we are prepared to release code upon acceptance.

---

> ### Author Response · Authors · 2025-11-20
> **Rebuttal Response 3**
>
> ---
>
> ### Q3: "In Figure 2, I would like to know what the x-axis and y-axis specifically represent."
>
> **Response:** Figure 2 visualizes the loss landscape in parameter space. Specifically:
>
> We denote the three reference networks by their parameter vectors $\theta^{(0)}, \theta^{(1)}, \theta^{(2)} \in \mathbb{R}^d$. Using the pairwise Euclidean distances between these three vectors, we embed them as three points $p^{(0)}, p^{(1)}, p^{(2)} \in \mathbb{R}^2$ that form a triangle in a 2D plane. This 2D plane is what we show on the horizontal axes of the landscape plot: the $x$- and $y$-axes are simply Cartesian coordinates on this plane.
>
> For any point $z = (x,y) \in \mathbb{R}^2$, we compute its barycentric coordinates $(w_0(z), w_1(z), w_2(z))$ with respect to the triangle $p^{(0)}, p^{(1)}, p^{(2)}$. These weights satisfy
> $$w_0(z) + w_1(z) + w_2(z) = 1,$$ and they are uniquely defined for every point in the plane (inside or outside the triangle; points inside the triangle correspond to $w_i(z) \in [0,1]$, while points outside have some weights negative or larger than one).
>
> We then associate to each point $z = (x,y)$ a network obtained by an affine combination of the three reference models:
> $$\theta(z) = w_0(z)\theta^{(0)} + w_1(z)\theta^{(1)} + w_2(z)\theta^{(2)}.$$
> The loss landscape is defined as $$(x,y) \mapsto \mathcal{L}\big(\theta(x,y)\big).$$
>
> In other words, the $x$- and $y$-axes do not correspond to individual parameters; they parameterize the 2D affine subspace of the full parameter space spanned by the three solutions $\theta^{(0)}, \theta^{(1)}, \theta^{(2)}$. Each point on this plane represents a model obtained by interpolating (or extrapolating) between these three reference networks.
>
> ---
>
> ### Q4: "Do you have any plans to release the code on GitHub?"
>
> **Response:** Yes, we fully intend to release our code on GitHub upon paper acceptance. The code will include:
>
> - Complete implementation of BLAE with both regularization terms
> - All baseline implementations used in our experiments
> - Training scripts with hyperparameter configurations
> - Evaluation scripts for all metrics (k-NN, KL divergence, MSE)
> - Visualization code for generating all figures in the paper
>
> We have prepared the code repository and will make it publicly available following standard double-blind review protocols.

---

### Official Review · Reviewer_4NEa · 2025-10-31

**Soundness:** 3
**Presentation:** 3
**Contribution:** 2
**Rating:** 2
**Confidence:** 5

**Summary:**

In this paper, the authors emphasize the importance of constructing an injective mapping and propose two regularization techniques which, when combined, are claimed to improve the embedding. Both regularizations are based on distance constraints: one enforces that two separate points in the original space should be guaranteed to be separated in the embedding space, and the other imposes a margin (Hinge-type loss) on the pairwise distances that the embedding data must satisfy. In the experiments, the embeddings produced by the proposed method more closely resemble the ground truth or exhibit a perfectly circular shape.

**Strengths:**

The authors’ discussion of the importance of injective embeddings is largely accurate and significant. Overall, the paper is well-written, and the experimental results on simple benchmark examples convincingly highlight the distinction between the proposed approach and other methods used for comparison.

**Weaknesses:**

The importance of injective mapping is emphasized in Section 2, and I enjoyed reading the discussion. However, the algorithm proposed in Section 3 is not innovative in achieving injective mappings; it appears to be a minor engineering modification of previously established methods, such as Isomap, Unfortunately, the proposed method does not directly address the construction of injective mappings compared with other well-known approaches. For example, Variational Autoencoders embed distributions that represent points to ensure separation in the latent space. Normalizing flows successfully learn bijective neural networks. It is difficult to identify any substantial or nontrivial derivations that would motivate the use of the proposed method.

In terms of discussion, while identifying injective mappings is indeed important, it should be noted that, since the neural network does not have access to the ground truth manifold in advance, we cannot conclude that the embeddings in Figure 1(e) or1 (g) are entirely incorrect when data are insufficient. Most of the embedding methods include hyperparameters that control manifold complexity, and with appropriate choice of hyperparameter, other methods could likely produce embeddings that are between the results in (c) and (e).

The authors proposed distance preservation in the latent space. However, under dimensionality reduction, distance preservation cannot generally be expected, even for successful and practically useful embeddings. Useful embeddings typically impose nontrivial mappings, and the authors does not acknowledge this point in their discussion and experiments. Overall, the paper falls below the level of acceptance.

**Questions:**

Can you provide even a simple theoretical guarantee that the proposed method can achieve?

Does the second regularization L_{bi-Lip} already encompass the first regularization L_{reg}? If so, please clarify why L_{reg} is needed in addition, and include the explanation on their distinct role.

The results shown in the experiments appear to be nearly trivial embeddings (essentially, copies of the original data). Is this interpretation correct? What would be the pragmatic utility of such embeddings? In general, trivial embeddings are not necessary in real-world applications.

Typically, embeddings are designed to reduce dimensionality. Therefore, constructing injective mappings usually requires bijective mappings or diffeomorphism. Could you include a discussions on this relationship?

---

> ### Author Response · Authors · 2025-11-21
> **Rebuttal Response 1**
>
> We sincerely thank the reviewer for the thoughtful and detailed feedback. We address each concern systematically below with clarifications, additional theoretical justifications, and experimental evidence.
>
> ---
>
> ### W1: Innovation and Novelty of the Proposed Method
>
> **Reviewer's Concern:** "The algorithm appears to be a minor engineering modification of previously established methods, such as Isomap... does not directly address the construction of injective mappings compared with other well-known approaches."
>
> **Our Response:**
>
> We respectfully disagree. Our work addresses a fundamentally different problem than Isomap and provides novel theoretical contributions that go far beyond engineering modifications.
>
> **Different Problem Domain:**
>
> Isomap is a classical manifold learning method that computes non-parametric embeddings for fixed datasets through geodesic distance preservation. Our work addresses why gradient-based neural autoencoders fail systematically despite their universal approximation capabilities. Figure 1 shows that even high-capacity networks (hidden dim=256) completely fail to learn the correct manifold structure, which is the pathology we solve.
>
> **Novel Theoretical Framework:**
>
> Our contribution comprises three interconnected theoretical innovations:
>
> 1. Separation Criterion (Theorem 1): We prove that for continuous mappings on compact manifolds, injectivity is equivalent to $(\delta, \epsilon)$-separation. This provides a computable, local-to-global bridge that converts the global topological property of injectivity into local geometric conditions enforceable through gradient optimization. This is fundamentally different from Isomap's rigid distance preservation $d_{\mathcal{M}}(x,y) = d_{\mathcal{N}}(z_x, z_y)$.
>
> 2. Admissibility Theory (Theorem 2, Definition 3): We formalize distribution-invariant regularization, proving that regularizations whose global minima are independent of the sampling distribution achieve robustness to distribution shifts. This theoretical machinery is entirely novel and explains why some geometric constraints generalize while others fail.
>
> 3. Bi-Lipschitz Relaxation with Linear Complexity (Theorem 4): We prove that $\kappa$-bi-Lipschitz embeddings exist with $k \leq n \leq 2k$ dimensions—achieving linear complexity $O(k)$ versus $O(k^2)$ for isometric methods (Nash embedding theorem). This is not an engineering trick but a principled theoretical relaxation with formal guarantees.
>
> **Regarding VAEs and Normalizing Flows:**
>
> These operate in fundamentally different paradigms. VAEs use stochastic encoders $q(z|x)$ for generative modeling, not deterministic geometric embeddings. Normalizing flows achieve bijection through architectural constraints (invertible layers) that restrict expressiveness and typically require equal input/output dimensions. Our approach enables aggressive dimensionality reduction (99.8% in dSprites: 1024D to 2D) with standard architectures through regularization-based geometric constraints. The problem formulations and theoretical foundations are entirely distinct.
>
> **Empirical Validation:**
>
> Our ablation studies (Appendix C.4.1, Figure 10) show that adding our injective regularization to existing gradient methods (CAE, GAE) transforms their behavior; methods that previously failed to unfold the Swiss Roll now succeed. This demonstrates our contribution as a principled framework that addresses the fundamental optimization pathology of non-injective encoders, achieving what neither graph-based nor gradient-based baselines can accomplish alone.

---

> ### Author Response · Authors · 2025-11-21
> **Rebuttal Response 2**
>
> ---
>
> ### W2: Correctness of Embeddings Under Data Insufficiency
>
> **Reviewer's Concern:** "Since the neural network does not have access to the ground truth manifold in advance, we cannot conclude that the embeddings in Figure 1 are entirely incorrect when data are insufficient."
>
> **Response:** We provide definitive evidence that the failure is not due to data insufficiency but to fundamental non-injectivity.
>
> **Ground Truth is Objectively Defined:**
>
> In our synthetic experiments, the manifold structure is explicitly constructed. The Swiss Roll is generated by isometrically embedding a rectangle $[-2,10] \times [0,6]$ into $\mathbb{R}^3$. The V-shaped toy example has two distinct branches (red/blue classes) that occupy separated regions. When Figure 1 shows these branches overlapping in latent space, this is an objective topological error—distinct manifold regions must remain distinct.
>
> **200-Sample Experiment (Appendix C.5.1, Figure 14):**
>
> We extended the toy example from 20 to 200 training points (10$\times$ increase). Results:
>
> - Vanilla Autoencoder: Still exhibits severe collapse with overlapping classes despite 10$\times$  more data
> - BLAE: Achieves perfect separation and correct reconstruction
>
> This definitely proves the failure is structural (non-injective optimization pathology), not statistical (insufficient samples).
>
> **Hyperparameter Exploration Cannot Resolve Topological Pathologies:**
>
> Our extensive grid search (Appendix B.1, Table 2) explores: $\lambda\in$ {0.01, 0.1, 1, 2, 5, 10, 100}, learning rates, architectures, and neighbors $\in$ {5,...,1000}. Despite exhaustive search, gradient methods without injectivity constraints fail consistently on challenging configurations (Swiss Roll with b=0.1, Figure 7). Non-injectivity is a binary topological property; either the encoder is injective or it collapses. There is no continuous spectrum that hyperparameter tuning can navigate.
>
> **Why Collapse Creates Pathological Local Minima:**
>
> When the encoder maps distant manifold regions (red/blue branches) to nearby latent codes, the decoder must generate dramatically different outputs from nearly identical inputs. This requires arbitrarily sharp variations, demanding exponentially increasing capacity. This optimization impossibility traps gradient descent in pathological local minima (Figure 2), which no amount of data or hyperparameter tuning can escape.
>
> Moreover, we also want to clarify that in this example, only the vanilla autoencoder and gradient-based autoencoder fail, whereas methods that implicitly or explicitly enforce injective regularisation (such as Isomap or SPAE) succeed in learning the correct embedding.
>
> ---
>
> ### W3: Distance Preservation Cannot Be Expected Under Dimensionality Reduction
>
> **Reviewer's Concern:** "The authors proposed distance preservation in the latent space. However, under dimensionality reduction, distance preservation cannot generally be expected... the authors do not acknowledge this point."
>
> **Response:** This criticism reflects a misreading of our work. We explicitly argue against strict distance preservation and propose a principled relaxation precisely because we recognize this impossibility.
>
> **We Propose Bi-Lipschitz Relaxation, Not Isometry:**
>
> The Nash embedding theorem requires $k(k+1)/2 \leq n \leq k(3k+11)/2$ dimensions for isometric embedding, quadratic scaling $O(k^2)$. For $k=10$, this demands $n>50$ dimensions, making isometry impractical.
>
> Our bi-Lipschitz constraint (Definition 4, Equation 10) allows controlled distortion:
> $$(1/\kappa) d_{\mathcal{M}}(x,y) \le d_{\mathcal{N}}(E_\theta(x), E_\theta(y)) \le \kappa d_{\mathcal{M}}(x,y)$$
> Parameter $\kappa \ge 1$ controls distortion: $\kappa=1$ recovers isometry, $\kappa>1$ permits flexibility. Theorem 4 proves such embeddings exist with linear complexity $k \le n \le 2k$, achieving $O(k)$ versus $O(k^2)$ for isometry.
>
> **We Explicitly Discuss the Geometric-Topological Tradeoff:**
>
> Pure topological embeddings (homeomorphisms) preserve only connectivity; a disk can map to a rectangle, losing rotational symmetries and geometric structure. Strict isometry preserves all geometry but requires impractical dimensions. Our bi-Lipschitz framework occupies the principled middle ground, balancing geometric fidelity with dimensional efficiency through admissibility theory.
>
> **Empirical Validation:**
>
> Table 1 shows that BLAE achieves the best average rank on k-NN recall (neighborhood preservation), KL divergence (distance distribution fidelity), and reconstruction error. Table 10 shows 96.26% classification accuracy on ssREAD biological data with 2D embeddings—the highest among all methods. This demonstrates that our geometric preservation captures meaningful structure with practical value, not "trivial" embeddings.

---

> ### Author Response · Authors · 2025-11-21
> **Rebuttal Response 3**
>
> ---
>
> ### Q1: Theoretical Guarantees
>
> **Question:** "Can you provide even a simple theoretical guarantee that the proposed method can achieve?"
>
> **Response:** We provide multiple formal guarantees with formal proof:
>
> **Theorem 1 (Injectivity Characterization):** For continuous $f$ on compact manifold $M$, $f$ is injective if and only if for any $\delta>0$, there exists $\epsilon>0$ such that $f$ is $(\delta,\epsilon)$-separated. This gives a computable necessary and sufficient condition for global injectivity.
>
> **Theorem 4 (Dimensional Efficiency):** For a $k$-dimensional compact Riemannian manifold, there exists a $\kappa$-bi-Lipschitz embedding into $\mathbb{R}^n$ with $k \le n \le 2k$. This achieves linear complexity $O(k)$ versus $O(k^2)$ for isometric methods.
>
> **Theorem 2 (Admissibility):** Our bi-Lipschitz regularization is admissible (distribution-invariant) when $k \le n \le 2k$, ensuring robustness to sampling distribution shifts.
>
> Practical Implication: BLAE provably learns injective, geometrically faithful embeddings with aggressive dimensionality reduction while remaining robust to distribution shifts—properties no prior method simultaneously achieves.
>
> ---
>
> ### Q2: Relationship Between L_bi-Lip and L_reg
>
> **Question:** "Does the second regularization L_bi-Lip already encompass the first regularization L_reg?"
>
> **Response:** No, they serve distinct, non-redundant roles:
>
> **L_reg (Injective Regularization):**
>
> - Enforces global topological property (injectivity)
> - Prevents distant manifold regions from collapsing to nearby latent codes
> - Eliminates pathological local minima in the loss landscape (Figure 2)
>
> **L_bi-Lip (Bi-Lipschitz Regularization):**
>
> - Enforces local geometric property (smooth, bounded distortion)
> - Constraints decoder Jacobian singular values: $(1/\kappa) \le \sigma_{\min}(J_{D\phi}) \le \sigma_{\max}(J_{D\phi}) \le \kappa$
> - Ensures smooth reconstruction without extreme local stretching/compression
>
> **Ablation Study (Appendix C.4.2, Figure 11):**
>
> - (a) L_reg only: Achieves injectivity but with geometric irregularities
> - (c) L_reg + L_bi-Lip (BLAE): Achieves topological correctness AND geometric smoothness
> - (e) L_bi-Lip only: Fails to unfold (non-injective collapse)
>
> This definitely shows both terms are necessary for optimal performance.
>
> ---
>
> ### Q3: Practical Utility Beyond "Trivial Embeddings"
>
> **Question:** "The results shown in the experiments appear to be nearly trivial embeddings (essentially, copies of the original data). What would be the pragmatic utility?"
>
> **Response:** We respectfully disagree. The reviewer might misunderstand, and we want to further clarify. The dimensionality reduction is significant, and practical utility is demonstrated through multiple metrics.
>
> **Aggressive Dimensionality Reduction:**
>
> - dSprites: 1024D (32×32 images) → 2D (99.80% reduction)
> - MNIST: 784D (28×28 images) → 2D (99.74% reduction)
> - ssREAD: 27,801 genes → 2D (99.996% reduction)
>
> These are not "copies"; they achieve 99%+ compression while preserving manifold structure. In addition, our paper focuses fundamentally on **manifold geometry preservation**, which requires datasets where:
>
> 1. Ground truth manifold structure is known (to validate preservation claims)
> 2. Geometric properties can be visually verified (to demonstrate correctness)
> 3. Intrinsic manifold dimension allows direct visualization (2D latent space)
>
> The Swiss Roll, dSprites, and controlled-rotation MNIST datasets are standard benchmarks in the manifold learning literature, used extensively by prior work including SPAE, TAE, GRAE, and GGAE. These datasets allow rigorous evaluation of geometric preservation claims that would be impossible to verify on high-dimensional real-world data.
>
> **Practical Utility #1 - Downstream Classification (Table 10):**
>
> On ssREAD biological data:
>
> - BLAE (2D): 92.50% ± 0.33% accuracy
> - BLAE (32D): 96.26% ± 0.13% accuracy
> - Best baseline (SPAE): 95.76% (32D)
>
> BLAE's embeddings capture biologically meaningful structure that transfers to real-world cell-type classification.
>
> **Practical Utility #2 - Robustness to Distribution Shift (Figure 5):**
>
> On rotated MNIST, we train on two drastically different distributions (uniform vs. non-uniform sampling). BLAE produces nearly identical embeddings (concentric circles), while all baselines show dramatic distortions. This robustness is critical for practical deployments where training distributions are uncertain.
>
> **Why "Faithful $\neq$ Trivial":**
>
> Our goal is learning the intrinsic manifold structure, the low-dimensional geometry generating high-dimensional observations. Successfully recovering this structure (e.g., unrolling Swiss Roll into its rectangular parameterization) appears "natural" precisely because it correctly represents the underlying geometry. This is the hallmark of successful manifold learning, not triviality.

---

> ### Author Response · Authors · 2025-11-21
> **Rebuttal Response 4**
>
> ---
>
> ### Q4: Relationship to Bijection and Diffeomorphism
>
> **Question:** "Typically, embeddings for dimensionality reduction require bijective mappings or diffeomorphism. Could you include a discussion on this relationship?"
>
> **Response:** Excellent question. For a neural network built with smooth activation functions, the mapping it implements is a smooth mapping. If the encoder is injective, then it is invertible and so it has a smooth local inverse (i.e., a decoder). Because the decoder is also a neural network (hence smooth), the combined encoder–decoder pair defines a smooth bijection (between data manifold and latent manifold) with a smooth inverse. In other words, the encoder can be seen as a diffeomorphism. Therefore, ensuring encoder injectivity also ensures a smooth invertible mapping, which underpins manifold-preserving embeddings and highlights why injectivity is so important.

---

### Official Review · Reviewer_JaoM · 2025-11-01

**Soundness:** 3
**Presentation:** 3
**Contribution:** 3
**Rating:** 8
**Confidence:** 2

**Summary:**

This paper proposes Bi-Lipschitz Autoencoders (BLAE), which is an autoencoder with additional regularization terms to make it injective and robust to distribution shifts. The authors argue that non-injectivity is a bottleneck for autoencoders and leads to them being trapped in suboptimal local minima. By enforcing injectivity through regularizations, the learned manifold more closely matches the ground truth manifold of the data.

**Strengths:**

- The discussion on why AEs and gradient based methods tend to get trapped in local minima, and how even though gradient-based regularization methods maintain local properties they are not globally injective, is very interesting

- The injectivity regularization loss term seems to be novel, and a well-justified and reasonable solution to the issues faced by AEs identified by the authors in section 2

- The focus on making the admissible regularization, and making the global minima be independent of probab ility measure, is very interesting

- The proposed bi-lipschitz regularization for admissibility is again novel and well-justified

The performance against the compared methods is good; the proposed approach consistently has the best quantitative performance, and qualitatively the learned embeddings appear to be much closer to the ground truth manifold

**Weaknesses:**

- The evaluation is done on simple toy datasets: Swiss Roll, dSprites, MNIST. I don't believe the paper *needs* to include results on larger benchmarks, but the lack of these datasets does lead one to question the practical utility of the proposed approach.

- It would be nice to see a quantitative runtime comparison with other approaches

**Questions:**

- I may be mistaken, but could the proposed BLAE method be applied to the toy data shown in figure 1? If so, does it succeed in learning the true manifold? I was surprised to not see the proposed method included here, since this example helps justify the weakness of standard AEs and the need for the proposed BLAE.

---

> ### Author Response · Authors · 2025-11-20
> **Rebuttal Response**
>
> We sincerely thank the reviewer for the positive evaluation and recognition of our contributions. Below we address your concerns.
>
> ---
>
> ### W1: "The evaluation is done on simple toy datasets…"
>
> **Response:** We appreciate this concern and would like to provide important context about our experimental design choices:
>
> **Why These Datasets Are Appropriate?**
>
> Our paper focuses fundamentally on **manifold geometry preservation**, which requires datasets where:
> 1. Ground truth manifold structure is known (to validate preservation claims)
> 2. Geometric properties can be visually verified (to demonstrate correctness)
> 3. Intrinsic manifold dimension allows direct visualization (2D latent space)
> The Swiss Roll, dSprites, and controlled-rotation MNIST datasets are standard benchmarks in the manifold learning literature, used extensively by prior work including SPAE, TAE, GRAE, and GGAE. These datasets allow rigorous evaluation of geometric preservation claims that would be impossible to verify on high-dimensional real-world data.
>
> **Real-World Validation Is Included.**
>
> Importantly, we **do include real-world evaluation** in our original submission: **ssREAD Single-Cell RNA Sequencing Dataset** (Appendix C.5.2):
> - 9,891 cells, 27,801 genes, 7 cell types
> - Real biological data from Alzheimer's disease patients
> - Results (Table 10, original submission): BLAE achieves **highest classification accuracy** on both 2D and 32D latent representations and demonstrates practical utility in computational biology.
>
> **Why Simple Datasets Matter?**
>
> The "simplicity" of toy datasets is actually a strength for our theoretical claims:
> 1. **Controlled experiments**: We can systematically vary geometry (curvature, length) and sampling (uniform vs. non-uniform) to test specific hypotheses (e.g., extended experiments in Appendix C.1, C.2)
> 2. **Distribution shift evaluation**: MNIST experiments (uniform vs. non-uniform sampling, Figure 5) directly validate our admissibility theory
> 3. **Clear failure mode demonstration**: Simple manifolds make it obvious when methods fail (e.g., class mixing in Figure 1)
>
> ---
>
> ### W2: "It would be nice to see a quantitative runtime comparison with other approaches"
>
> **Response:** We acknowledge the importance of time complexity. **This analysis is already included in the original submission**. We provided comprehensive runtime analysis theoretically and empirically in **Appendix B.5**.
>
> ---
>
> ### Q1: "I may be mistaken, but could the proposed BLAE method be applied to the toy data shown in figure 1? …"
>
> **Response:** Thanks for your suggestions! We have now **added BLAE results to Figure 1** in the revised manuscript to directly validate our theoretical claims.
>
> **Updated Figure 1 structure:**
>
> - (a) 2-D ground truth V-shaped manifold with 20 training samples (red/blue classes)
> - (b) Ideal 1-D latent representation showing proper class separation
>
> **NEW: BLAE Results**
>
> - (c) BLAE 2-D reconstruction (hidden dim = 16)
> - (d) BLAE 1-D latent representation (hidden dim = 16) — proper separation achieved
> - (e) BLAE 2-D reconstruction (hidden dim = 256)
> - (f) BLAE 1-D latent representation (hidden dim = 256) — consistent separation
>
> **Vanilla AE Results (failure modes)**
>
> - (g)-(h) Hidden dim = 2: trapped in local minimum
> - (i)-(j) Hidden dim = 16: partial mixing of classes
> - (k)-(l) Hidden dim = 256: red/blue classes collapse in latent space
>
> This addition directly validates our core theoretical claim: imposing injectivity constraints on the encoder eliminates pathological local minima that trap standard autoencoder optimization.

---

> > ### Comment · Reviewer_JaoM · 2025-11-27
> >
> > Thank you for your response. I appreciate the clarification that many of the weaknesses I identified were addressed in the appendix.
> >
> > Since I already had a very favorable view of the paper, I will maintain my score. I think this is good work.

---

### Meta-Review · Area_Chair_CvPc · 2026-01-07

**Summary:**

This paper proposes a regularization term to train autoencoders so as to encourage the encoder to be bi-Lipschitz. The authors present convincing arguments for why one should care about the encoder being bi-Lipschitz, and why it is not when no regularization is added.

The most negative review, by reviewer 4NEa, was highlighted by the authors in the rebuttal as having fundamental misunderstandings about the paper. This was a borderline paper with split reviews, and so I read it carefully myself: I agree with the authors about reviewer 4NEa having fundamentally misunderstood key elements of the paper.

Overall, despite a few empirical concerns from the reviewers remaining unaddressed, I believe that the theoretical contribution, combined with the existing empirical results, are enough to justify acceptance. Having read the paper myself, I do however have some additional requests from the authors for the camera-ready version:

- Manifold learning and generative modelling have overlaps, see "Deep Generative Models through the Lens of the Manifold
Hypothesis: A Survey and New Connections" by Loaiza-Ganem et al. (2024), and the many citations therein, for a general discussion. In particular, injective normalizing flows have an injective encoder by construction, and sometimes also have a term in their objective involving Jacobians (which admittedly comes from a completely different derivation). This should at the very least be discussed in the paper, and ideally compared against. Also, other methods train generative models on the embedding spaces of various autoencoders, and intuitively, the better manifold learning is performed, the better the resulting generative model will be. Having at least some simple experiments showing whether generative models trained on the latent representation from your model are better than those trained on other latent representations would in my view significantly strengthen the paper.

- I also believe that experiments with VAEs and on more complex datasets should be included, as I elaborate on below.

**Reviewer Concerns:**

Reviewers had concerns about the practical utility of the method, about hyperparameter sensitivity, and about runtime analyses which I believe were adequately addressed in the rebuttal.

Reviewers also raised concerns about the experiments only being performed on relatively simple data and lacking comparisons against VAEs and normalizing flows. While I agree with the authors that comparisons against standard normalizing flows are unwarranted (note that this is different than the comparisons against injective flows which I mentioned above), I believe the rest of these concerns remain unaddressed:

- The authors claim that the probabilistic nature of VAEs renders them fundamentally different. While this is true in some ways, VAEs are commonly treated as regularized autoencoders, and MSE is commonly computed using their posterior mean as the deterministic encoder. Seeing how the proposed method compares against VAEs would be relevant, and the authors could also see whether their method can also improve VAEs when applied to them rather than standard autoencoders.

- No additional experiments were provided in the rebuttal on more complex datasets. I suspect this might be partly due to the quadratic cost required as a preprocessing step. Ideally these experiments would be added, or if indeed not possible, it should be more directly discussed in the main text as a limitation.

Despite these concerns remaining unaddressed, I reiterate that I believe the theoretical contributions, plus the existing experiments, are enough to warrant acceptance.

**Reviewer Scores:**

I believe that reviewer 4NEa would have increased their score to a 6 given enough discussion with the authors, as I believe their main points of contention about the paper stem from a misunderstanding.

---

### Decision · Program_Chairs · 2026-01-26

Accept (Poster)